# Twenty-year trends in antimicrobial resistance from aquaculture and fisheries in Asia

Daniel Schar [1]✉, Cheng Zhao [2], Yu Wang [2], D. G. Joakim Larsson [3,4], Marius Gilbert[1,5,7] & Thomas P. Van Boeckel [2,6,7]✉

Antimicrobial resistance (AMR) is a growing threat to human and animal health. However, in aquatic animals—the fastest growing food animal sector globally—AMR trends are seldom documented, particularly in Asia, which contributes two-thirds of global food fish production. Here, we present a systematic review and meta-analysis of 749 point prevalence surveys reporting antibiotic-resistant bacteria from aquatic food animals in Asia, extracted from 343 articles published in 2000–2019. We find concerning levels of resistance to medically important antimicrobials in foodborne pathogens. In aquaculture, the percentage of antimicrobial compounds per survey with resistance exceeding 50% (P50) plateaued at 33% [95% confidence interval (CI) 28 to 37%] between 2000 and 2018. In fisheries, P50 decreased from 52% [95% CI 39 to 65%] to 22% [95% CI 14 to 30%]. We map AMR at 10-kilometer resolution, finding resistance hotspots along Asia's major river systems and coastal waters of China and India. Regions benefitting most from future surveillance efforts are eastern China and India. Scaling up surveillance to strengthen epidemiological evidence on AMR and inform aquaculture and fisheries interventions is needed to mitigate the impact of AMR globally.

[1] Spatial Epidemiology Laboratory, Université Libre de Bruxelles, Brussels, Belgium. [2] Institute for Environmental Decisions, ETH Zurich, Zurich, Switzerland. [3] Center for Antibiotic Resistance Research, University of Gothenburg, Gothenburg, Sweden. [4] Department of Infectious Diseases, Institute for Biomedicine, University of Gothenburg, Gothenburg, Sweden. [5] Fonds National de la Recherche Scientifique, Brussels, Belgium. [6] Center for Diseases Dynamics, Economics, and Policy, New Delhi, India. [7] These authors jointly supervised this work: Marius Gilbert, Thomas P. Van Boeckel. ✉email: dlschar@gmail.com; thomas.vanboeckel@env.ethz.ch

A quaculture and fisheries contribute a growing share of nutrition for the global population[1]. Aquatic animals provide 20% of animal protein to the human diet for over 40% of the world, with consumption growth outpacing rates for all other sources of animal protein combined[1]. Driven by increasing demand, global fish production is experiencing rapid growth. From 1960 to 2018, aquatic animal production for human consumption increased from 21.8 to 156.4 million tons[1,2]. Asia contributes the largest share—69% in 2018—with China alone representing 35% of global production[1].

As capture fisheries production has plateaued since the early 1990s, aquaculture production has risen commensurate with global demand and now accounts for the majority (52%) of aquatic food animal production with continued expansion expected through 2030[1]. Growth in cultured aquatic animals averaged 5.3% annually since 2001 with select countries—notably Indonesia and Bangladesh—exceeding 9% annual growth.

The rapid growth in animal protein production has been facilitated by a transition from extensive to intensive farming, which in terrestrial food animal sectors has historically been accompanied by the increasing use of antimicrobials[3,4]. As a result of this global shift in animal production, and growing demand for animal-source foods, the terrestrial and aquatic food animal production industries have emerged as the largest consumer by volume (73.7% and 5.7%, respectively) of antimicrobials globally[3–5]. In aquaculture, some species of fish, such as catfish, are associated with antimicrobial use rates per kilogram that exceed those in terrestrial animals and humans[5].

Antimicrobial use exerts selective pressures driving antimicrobial resistance. In terrestrial animals, a growing body of evidence has linked AMR with productivity loss and resistant infections carrying harmful consequences to animal and human health[6–9]. In aquaculture and fisheries—although more limited than in terrestrial animals—the evidence concerning antimicrobial resistance has also expanded over the last decade[10–13]. Documenting the movement of resistance determinants in aquatic settings presents challenges, and the body of evidence from which to draw conclusions on antimicrobial resistance transference between aquatic animals and humans remains very limited. Yet the aquatic food animal supply chain may be an under-appreciated route for transmission of antimicrobial-resistant bacteria and resistance genes from aquatic animals and their environment to humans[14–18]. Mobile genetic elements carrying resistance genes of human clinical significance have been associated with aquaculture and the aquatic environment[14,18,19]. Aquatic food animal supply chains are highly globalized[1], facilitating the distribution of locally generated resistance at a global scale[20,21]. In addition, compared with other animal source foods, aquatic animal products are more likely to be consumed raw, increasing the risk of pathogen transmission.

Rising AMR rates are expected to disproportionately affect low-income and middle-income countries, jeopardizing development gains in vulnerable communities, widening economic inequality, and contributing to a rise in extreme poverty by 2030[22]. And resistance in pathogens of production significance may reduce treatment options in commercial aquaculture, with potential implications for food security and nutrition[13,23,24].

Strengthening surveillance to guide AMR interventions is a challenge across sectors. This challenge is particularly acute in the aquaculture and fisheries industry. Presently, even the most heavily consumed aquatic animals globally—freshwater and marine fish—are generally not subject to systematic foodborne pathogen surveillance. Point prevalence surveys provide evidence at discrete geographic and temporal scales and have been used to characterize the global distribution and burden of infectious disease in humans[25] and AMR in terrestrial animals[26]. These surveys may serve as a surrogate in the absence of routine, systematic surveillance, collectively providing a mosaic portrait of antimicrobial resistance trends. Enhanced documentation of AMR trends could then inform targeted surveillance programs and interventions in the world's most productive aquaculture and fisheries region[20,23,27].

Here, we summarize current evidence on AMR in aquatic food animals in Asia using a systematic review of point prevalence surveys. We map AMR levels at 10 km resolution, provide a baseline to monitor future AMR trends, and identify regions where future surveillance efforts should be prioritized.

## Results

The systematic review identified 749 point prevalence surveys reporting antimicrobial resistance rates in aquatic food animals in Asia published between 2000 and 2019. We extracted 12,698 resistance rates representing 11,289 isolates and 45 bacterial genera (Supplementary Figs. S1 and S2; Supplementary Data 1). Eastern Asia accounted for 50.6% ($n = 379$) of surveys; Western and Southern Asia, 30.7% ($n = 230$); and South-eastern Asia, 18.7% ($n = 140$). China, India, and Turkey together contributed nearly two-thirds of all point prevalence surveys across Asia. China alone represented 37.9% ($n = 284$) of surveys, a fraction that expanded over the last decade. India and Turkey each contributed 12.5% ($n = 94$) of surveys (Supplementary Figs. S3–S5).

**AMR trends from aquaculture and wild-caught fisheries.** Between 2000 and 2018, the percentage of antimicrobial compounds with resistance exceeding 50% (P50) in each survey plateaued in cultured aquatic animals at 33% [95% confidence interval (CI) 28 to 37%], and decreased sharply in wild-caught aquatic animals from 52% [95% CI 39 to 65%] to 22% [95% CI 14 to 30%] ($p = 0.003$) (Fig. 1). Across all years, the median P50 of surveys from cultured aquatic animals (31%; $n = 558$) was lower than surveys from wild-caught aquatic animals (44%; $n = 81$) ($p = 0.059$) (Supplementary Fig. S7).

**AMR profiles in foodborne pathogens.** The five most frequently isolated bacteria genera identified in our review—*Vibrio*, *Aeromonas*, *Streptococcus*, *Edwardsiella*, and *Escherichia* (*E.coli*)—together accounted for 68.5% of surveys. *Vibrio* spp. ($n = 191$) and *Aeromonas* spp. ($n = 174$) contributed nearly half of all surveys.

In foodborne pathogens, we calculated the pooled prevalence of resistance from individual pathogen-drug resistance rates (see "Methods" section). Amongst these foodborne pathogens, resistance was highest to penicillins (60.4%), macrolides (34.2%), sulfonamides (32.9%), and tetracyclines (21.5%) (Fig. 2). Although highly variable, mean resistance rates to the highest priority critically important antimicrobials for human medicine[28] were highest for macrolides (34.2%, 95% CI 33 to 35%) followed by third-generation and fourth-generation cephalosporins (17.5%, 95% CI 17 to 18%) and quinolones (16.2%, 95% CI 15 to 17%). Mean resistance to third-generation cephalosporins in *E.coli* was 27.1% (95% CI 25 to 29%) across surveys. Resistance to compounds in the reserve group of last-resort antimicrobials for human medicine[29] varied by pathogen: fosfomycin resistance in *E.coli* was 10.3% (95% CI 6 to 14%); polymyxin B resistance in *E.coli* was 19.4% (95% CI 7 to 32%), in *Vibrio* spp. was 39.1% (95% CI 33 to 44%) and in *Aeromonas* spp. was 71.5% (95% CI 67 to 76%); and colistin resistance in *E.coli* was 5.2% (95% CI 2 to 8%), in *Vibrio* spp. was 42.7% (95% CI 38 to 47%) and in *Aeromonas* spp. was 51.5% (95% CI 46 to 57%). In Gram-negative bacteria across all surveys, the mean colistin resistance was 41.3% (95% CI 39 to 44%).

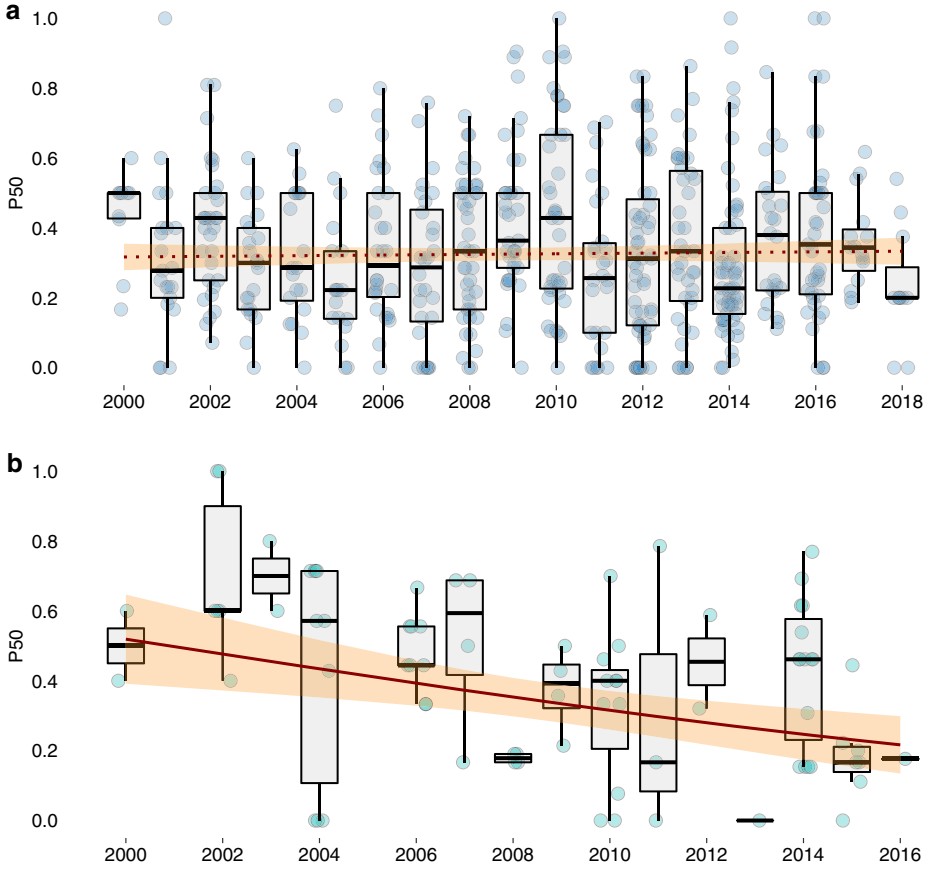

**Fig. 1 Annual trends in the proportion of drugs with resistance greater than 50% (P50) in each survey.** (**a**) P50 for cultured aquatic animals ($n = 558$); and (**b**) wild-caught aquatic animals ($n = 81$). The horizontal box lines represent the first quartile, the median, and the third quartile. Whiskers denote the range of points within the first quartile $-1.5\times$ the interquartile range and the third quartile $+1.5\times$ the interquartile range. Each survey is represented by a dot with horizontal jitter for visibility. Regression lines are fitted using generalized linear models, with a solid line indicating statistical significance ($p = 0.003$); 95% confidence intervals are shown in shaded areas.

Mean foodborne pathogen resistance to carbapenems was low (2.8%, 95% CI 2 to 4%), with the exception of *Aeromonas* spp. In *Aeromonas* spp. across all regions, carbapenem resistance increased from 5.1% (95% CI 2 to 8%) before 2010 to 51.1% (95% CI 43 to 60%) after 2010 ($p < 0.0001$). *Aeromonas* spp. in Western and Southern Asia exhibited elevated resistance compared with other subregions, particularly in samples originating from freshwater and marine fish. In these aquatic food animals across all years, mean carbapenem resistance in *Aeromonas* spp. was 40.3% (95% CI 33 to 48%), aztreonam resistance was 56.6% (95% CI 46 to 67%), and mean resistance to third-generation and fourth-generation cephalosporins was 69.6% (95% CI 65 to 75%).

**Geography of resistance.** Predicted hotspots (P50 > 0.5) of multi-drug resistance in freshwater environments included eastern Turkey; southern India—particularly the wetlands of coastal Kerala, Tamil Nadu, and Andhra Pradesh; the Yangtze River in China, both along its upper reaches and at Poyang Lake; and the lower reaches of the Mekong River and its delta in southern Cambodia and Vietnam. Low P50 (<0.1) was predicted in peri-urban Guangzhou in southern China and in South Korea and Japan (Fig. 3). The interpolation of resistance in this study is associated with uncertainty. Variability in the geolocation of surveys, in covariates, and in estimates of resistance contribute to this uncertainty, which is captured with a 95% confidence interval

map on P50 predictions (Supplementary Fig. S18). Uncertainty in P50 predictions was high (95% CI > 0.5) in South Korea and Japan.

We identified the locations of 50 hypothetical surveys to be conducted in Asia that would maximize information gained over the current map of AMR in freshwater environments. Future survey locations were optimized using a "need for surveillance" index, calculated as the uncertainty in AMR weighted to areas where resistance is likely to have the greatest impact on human health and the aquaculture industry (see "Methods" section). The majority of future surveys were projected in China (56%) and India (16%), with the highest tier of prioritization distributed predominantly in central and eastern China, western and central India, and along the Indo-Gangetic Plain (Fig. 4). Indonesia accounted for the third-highest number of surveys to be conducted in the future (12%).

In marine environments, predicted multi-drug resistance was highest along northeastern China on the Yellow and East China Seas; eastern Hainan Island waters in southern China; the coastal waters of central Vietnam on the South China Sea; southern India coastal waters on the Arabian Sea and the Bay of Bengal between southern India and northern Sri Lanka; and the eastern Mediterranean Sea on the coast of Lebanon. The coastal waters of Thailand and Malaysia on the Gulf of Thailand and Thailand's coastal waters on the Andaman Sea carried lower P50 predicted values (<0.3) (Fig. 5).

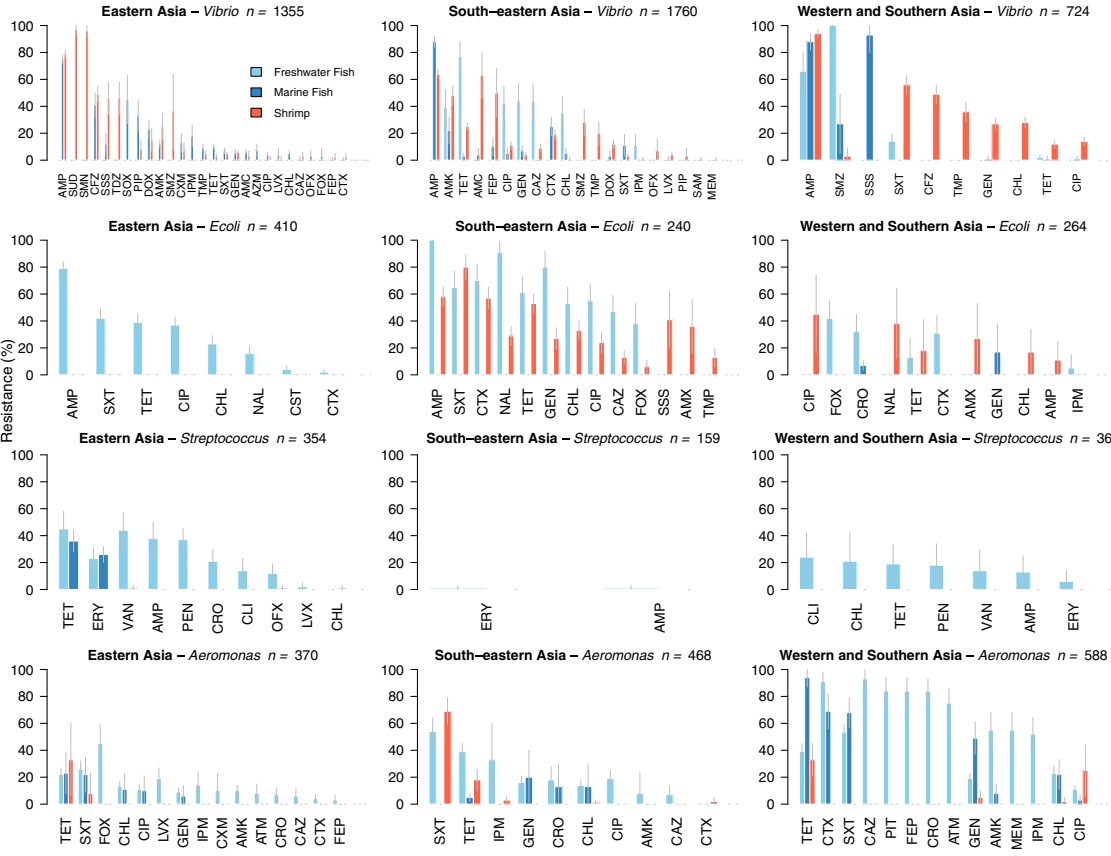

**Fig. 2 Antimicrobial resistance in foodborne pathogens isolated from aquatic animals in Asia.** Gray bars represent 95% proportion confidence intervals. Resistance is shown for pathogen-drug combinations recommended for susceptibility testing (Supplementary Table S1) and with 10 or more isolates tested. (For drug acronyms, see Supplementary Note 2).

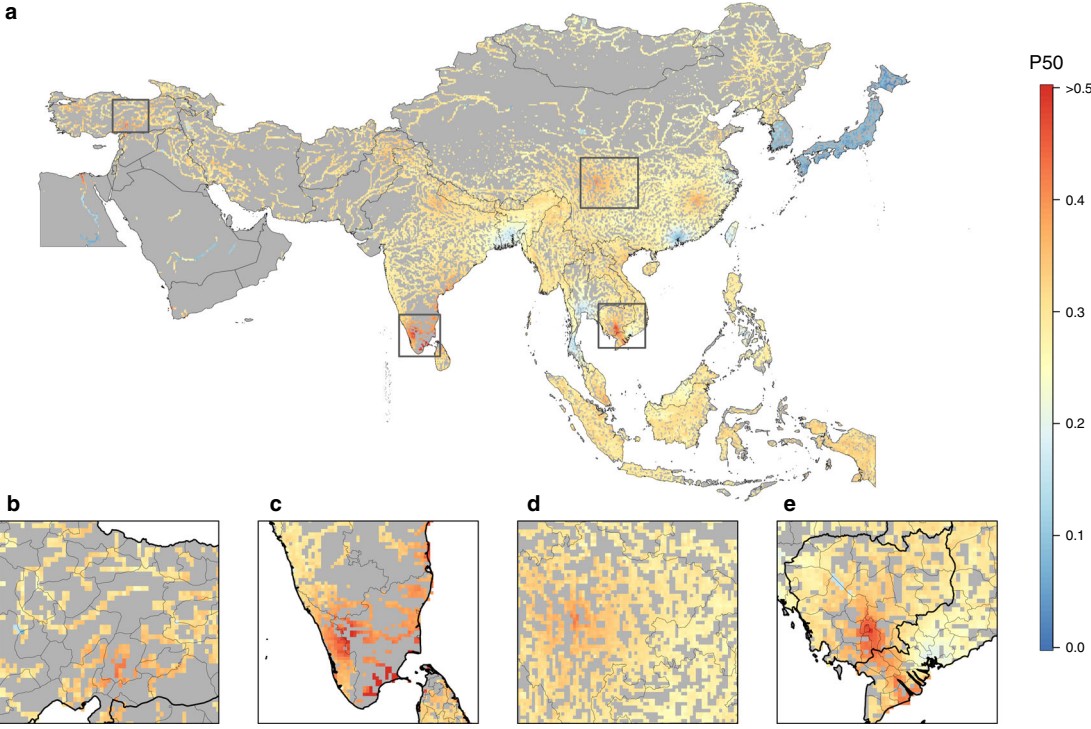

**Fig. 3 Geographic distribution of antimicrobial resistance in freshwater environments in Asia.** The proportion of antimicrobial compounds in each survey with resistance higher than 50% (P50) at continental scale (**a**); eastern Turkey (**b**); southern India (**c**); Yangtze River drainage basin in China (**d**); and the Mekong River delta (**e**).

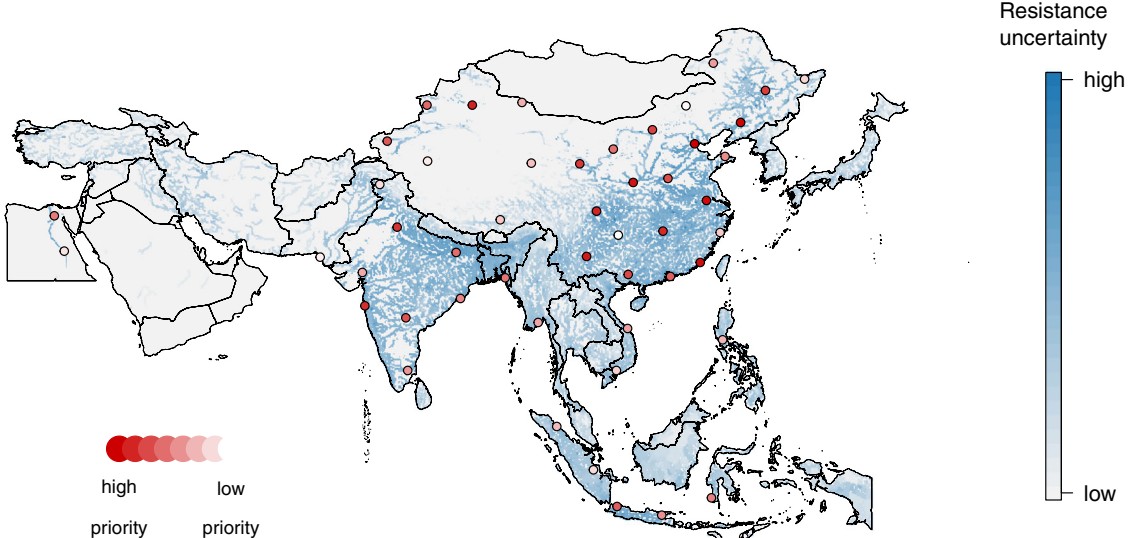

**Fig. 4 Future survey locations prioritized to reduce uncertainty in antimicrobial resistance in freshwater environments in Asia.** The background color gradient (blue) represents weighted uncertainty in multi-drug resistance (see "Methods" section). An initial set of 50 future surveys optimized to reduce uncertainty in multi-drug resistance is displayed (red).

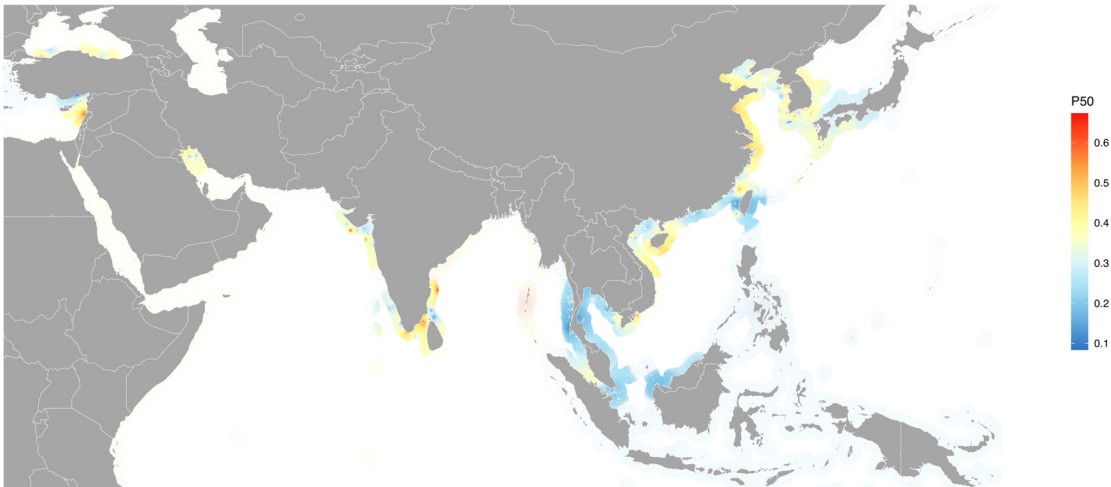

**Fig. 5 Antimicrobial resistance in marine environments in Asia.** Transparency reflects low survey density; areas of higher relative survey density are represented by increased opacity.

## Discussion

We reviewed and mapped antimicrobial resistance in aquatic food animals in Asia during a period of substantial industry growth. Our findings indicate that between 2000 and 2018, antimicrobial resistance in bacteria from cultured aquatic food animals was stable (33%) while the resistance from wild-caught aquatic food animals decreased sharply (52% to 22%). These trends represent currently available evidence from point prevalence surveys, which serve as a surrogate in the absence of systematic surveillance and should be interpreted cautiously. Structured, systematic surveillance will be imperative to document trends in multi-drug resistance at the sub-national level in the future.

Our results are consistent with an analysis of antimicrobial resistance in aquaculture-derived bacteria from forty countries, nearly half of which in Asia, which identified a global mean multi-antibiotic resistance index of .25, and a higher index (>.35) in low-income and middle-income countries in Asia[27]. Although antimicrobial use in surveys from cultured animals was most frequently unspecified, in the limited surveys that recorded whether on-farm antimicrobials were either used or not used ($n = 63$; 11%), use was associated with higher multi-drug resistance than the absence of use ($p < .001$) (Supplementary Fig. S8). The combined effect of multi-source antimicrobial introduction[15] and persistence[30,31] in aquaculture waters and sediments may present a consistent selective pressure facilitating the maintenance of an elevated level of antimicrobial resistance.

Declining resistance observed in bacteria from wild-caught aquatic animals could be associated with reduced human and livestock fecal pollution exposure in wild-caught animals over time. Anthropogenic influences such as human and livestock fecal pollution have been positively associated with antimicrobial resistance gene frequency[32,33], and in these surveys, we observed a negative correlation between multi-drug resistance and the World Bank basic sanitation index[34] (Supplementary Fig. S9). In addition, surveys from the first half of the study period held a significantly higher proportion of diseased animals than those from the latter half, and disease was positively associated with resistance (Supplementary Fig. S10). Importantly, these findings must be considered in the context of relatively scarce point

prevalence surveys (11%, $n = 81$) attributable to capture fisheries. Other contributions—including improved wastewater management and regulatory action—cannot be ruled out and future studies will be valuable in documenting resistance trends and their influencing factors in capture fisheries.

Temporal trends in P50 rates reflect resistance across multiple bacteria, aquatic animal species, production contexts, and sub-regions in Asia. Production dynamics, antimicrobial exposure intensity, and regulatory standards, amongst other factors, may be highly variable by location and at timescales shorter than the duration of our analysis. Consequently, these trends may not capture more granular country-level resistance profiles in select pathogen species from specific aquatic animal production settings (Supplementary Fig. S11).

Amongst foodborne pathogens, we observed a concerning dual-threat profile, marked by high rates of resistance to first-line antimicrobial classes and, for *Vibrio* and *Aeromonas* spp., moderate to high rates of resistance to antimicrobial classes of last-resort reserved for treatment of multi-drug resistant pathogens.

In Western and Southern Asia, *Aeromonas* spp. isolates exhibit high rates of resistance to tetracyclines, sulfonamides, aminoglycosides, monobactams, carbapenems, and cephalosporins, which, together with moderate resistance to fluoroquinolones, suggests therapeutic options for management of invasive or non-self-limiting *Aeromonas* infections may already be restricted in this subregion as compared with lower resistance profiles reported outside of Asia[35,36]. High rates of carbapenem resistance from *Aeromonas* spp. were also identified in this subregion. *Aeromonas* spp. produce inducible metallo-beta-lactamases, are inherently resistant to ampicillin, amoxicillin-clavulanate, and cefazolin, and are known to express a carbapenemase encoding gene (*cphA*) the significance of which remains unknown[37]. Recent work identified the fish pathogen *Aeromonas allosaccharophila* as the origin of mobile cephalosporinase genes, which confer resistance to beta-lactams, and may be associated with aquaculture antimicrobial use[38]. While beta-lactam antimicrobials are utilized in aquaculture[5,39], to the best of our knowledge, carbapenems are not used. Elevated carbapenem resistance identified in Western and Southern Asia may reflect inducible resistance under high environmental exposure to beta-lactam antimicrobials in this subregion, and, taken together with increasing carbapenem resistance in *Aeromonas* spp. recently reported from other regions[40,41], may warrant further investigation.

Although variable by subregion and foodborne pathogen, high rates of resistance to third-generation and fourth-generation cephalosporins were also observed. These findings are consistent with trends identified in terrestrial animal production[26]. Ceftriaxone, Ceftazidime, and Cefotaxime resistance in *E.coli* across subregions was 27.1% (95% CI 25 to 29%), but approached 40% in Western and Southern Asia and exceeded 50% in South-eastern Asia, suggesting that resistance to third-generation cephalosporins is established in at least some members of Enterobacteriaceae in Asia. Notably, resistance in *E.coli* must be interpreted in the context of human and terrestrial animal fecal pollution and may represent the serial accumulation of antimicrobial resistance genes as has been observed in some human pathogens such as *Shigella*[42]. Extended-spectrum beta-lactamase (ESBL)-producing Enterobacteriaceae capable of hydrolyzing third-generation cephalosporins are recognized as a serious threat to human health[43]. Our finding confirms reports[44,45] of an existing pool of ESBL genes that may be plasmid encoded[46] and raises the prospect of horizontal gene transfer conferring broader resistance within and across diverse bacterial genera with the consequent loss of this critically important collection of cephalosporins.

Elevated colistin resistance in Gram-negative bacteria from aquatic animals observed in this study mirrored high rates identified in terrestrial animals in Asia over a similar study period[26]. Colistin resistance across a breadth of bacterial genera in Asia may reflect both intrinsic and acquired resistance mechanisms. Intrinsic resistance to polymyxins in Gram-negative bacteria has been documented[47]. The possibility that a proportion of the colistin resistance identified is acquired cannot be ruled out, particularly through plasmid-mediated transfer amongst members of the Enterobacteriaceae family[48]. Such resistance may be attributable to the prolific use of colistin and other polymyxins in the region, which is only recently evolving through national regulatory action, and the possible recruitment of *mcr*-family mobile resistance determinants, their broad dissemination through horizontal gene transfer, and the wide distribution potential of aquatic environments[14,48,49]. Indeed, the distribution mechanics of AMR in aquatic systems under differing hydrological conditions is an expanding area of research that holds the promise of contributing to an enhanced understanding of AMR risk across compartments[50].

Previous work has shown that bacterial foodborne disease attributable to aquatic animal consumption contributes a non-negligible burden to human morbidity[51,52]. In this context, the high rates of resistance to multiple classes of critically important antimicrobials in aquatic animal foodborne pathogens in our study raise urgent concerns regarding both therapeutic efficacy of first-line antimicrobials and the further erosion of last-resort therapeutic options for multi-drug resistant infections resulting in severe disease.

Hotspots of AMR in freshwater environments were predicted along several of the region's great river systems, including the Yangtze River, and the estuaries of the Mekong and Nile Rivers. Low rates of AMR were predicted in Japan and South Korea. In these countries, the predictions were associated with high uncertainty (Supplementary Fig. S18). In our freshwater data set, South Korea contributed 2.4% of surveys and there were no surveys from Japan. Consequently, predictions in these two countries should be interpreted cautiously, requiring validation through additional surveillance. However, low rates of AMR could indicate less influence from human and livestock fecal pollution. Low AMR may also reflect comparatively lower rates of integrated livestock-aquaculture farming, enhanced environmental pollution regulatory capacities, and a heightened awareness of antimicrobial stewardship principles in these countries. Low AMR in the peri-urban environment of Guangzhou in southern China was unexpected considering the comparatively dense human population, prevalence of integrated livestock-aquaculture farming systems, and broad agricultural and human connectivity with the aquatic environment in Guangdong Province[14]. Future work will be important to corroborate these findings and identify factors that have the potential to moderate multi-drug resistance.

We identify 50 hypothetical future survey locations prioritized to maximize the knowledge gained over the current map of resistance from freshwater in Asia. Weighting the uncertainty in predicted resistance by the product of human population density and inland aquaculture production prioritizes future surveillance effort to locations where potential multi-drug resistance exposure risk and impact on human health and the aquaculture industry may be most significant (Supplementary Note 4). China and India together account for 72% of future surveys, reflecting their comparatively high population density and predominant role globally—first and second, respectively—in inland aquaculture production[2]. Our approach is scalable to individual country contexts and presents an opportunity to fill national or sub-national knowledge gaps and inform interventions and

stewardship under resource-limited settings by maximizing information gained through targeted surveillance.

In marine coastal environments, the highest rates of AMR were identified in northeastern China on the Yellow and East China Seas; southern China and central Vietnam on the South China Sea; southern India on the Arabian Sea and the Bay of Bengal between southern India and northern Sri Lanka; and the eastern Mediterranean Sea on the coast of Lebanon. Lower AMR rates were observed along Thailand and Malaysian coastal waters both on the Gulf of Thailand and the Andaman Sea. Aquaculture production has been correlated with aminoglycoside resistance gene abundance along estuarine waters of coastal China[53]. High rates of AMR in waters also identified as carrying the highest potential for marine aquaculture productivity[54], such as the coastal waters of Hainan Island on the South China Sea, may help prioritize surveillance in an industry poised for substantial growth. However, marine predicted values should be interpreted with caution, given the comparatively limited availability of marine surveys overall ($n = 322$) and an absence of surveys from countries such as Indonesia with sizeable marine aquaculture and nearshore marine fisheries industries.

As with any modeling study, our analyses come with limitations. First, point prevalence surveys, although numerous in the study ($n = 749$), present an inherent challenge to standardization of data from susceptibility testing that are subject to variability in methodologies and protocols resulting in uncertainties[55]. Second, the resistance across species within bacterial genera can vary markedly. Our analyses of drug resistance were conducted at the bacterial genera level—the most granular level at which complete data were available. Although analysis of trends at the bacterial species level may be valuable in context-specific settings, the aim of our study was rather to document temporal trends over twenty years in Asia across a broad range of bacteria. Third, our study includes samples originating from both healthy and diseased animals. An over-representation of samples from diseased animals associated with treatment failure could present a bias toward higher resistance rates[56]. Indeed, samples from diseased animals had higher median P50 values than samples from healthy animals. However, any influence on resistance across the study is likely moderated by a balanced mix between healthy (42%) and diseased (58%) animal samples from the surveys and our treatment of intermediately susceptible isolates as susceptible in our analysis. As drug tolerance precedes clinical resistance[57], it is possible that our results underestimate non-susceptibility. Fourth, the volume of data is currently insufficient to map AMR across different study periods using spatio-temporal interpolation methods. Overall, there was an increasing volume of surveys conducted in recent years (Supplementary Fig. S4). However, when evaluated by time series that most closely partitions each of the cultured and wild-caught data sets in half (cultured: 2000 to 2010 vs. 2011 to 2018; and wild-caught: 2000 to 2009 vs. 2010 to 2016), the directionality of the regressions remained unchanged, suggesting the increasing volume of surveys does not influence the temporal trends in P50. Fifth, due to the contiguous relationship between aquatic animals and their environmental waters, the interpretation of resistance in commensal isolates such as *E.coli* must be contextualized as indicative of human and livestock fecal contamination[32], either in origin waters or through post-harvest contamination. While characterizing the origin of selective pressures driving resistance is critical to risk mitigation at source, our study documents elevated resistance in bacteria from aquatic animals intended for human consumption—irrespective of origin—and can thus be interpreted as a risk to human health from contact with or consumption of aquatic animals and their products. Finally, it is notable that no surveys were available from Indonesia, despite contributing nearly 10% of global aquatic

animal production. Vietnam—the fourth largest global producer of cultured fish[1]—represented less than 3% of all surveys. An absence of surveys from Myanmar and Laos also illustrates gaps in understanding antimicrobial resistance in aquatic food animals in Asia. Despite our broad search parameters, surveys conducted in these countries and which were either not identified in our search or not available for review could potentially influence our findings. Similarly, in the absence of systematic surveillance, reliance on passive surveillance data presents variability in survey coverage, which could introduce bias, as well as uncertainty in survey geolocation, adding potential uncertainty to predictive models (Supplementary Fig. S18). Our study addresses this uneven geographic representation of point prevalence surveys by identifying—on an objective basis—sub-regions in Asia that would benefit most from further surveillance efforts. Such future surveillance will be essential to an enhanced and refined understanding of resistance trends in Asia.

This study identified elevated rates of antimicrobial resistance in bacteria isolated from aquatic animals intended for human consumption in Asia. A growing aquatic food animal production industry may serve as an important pathway for transmission of resistance along the food chain with potential consequences for human health.

A scale-up of an antimicrobial resistance surveillance architecture for aquatic food animals is urgently needed to fill gaps in AMR trends at national and sub-national levels. Our findings help direct the prioritization of this future surveillance effort and provide a foundation for establishing time-bound, measurable targets for reducing antimicrobial resistance[58,59]. The spatial profile of antimicrobial resistance presented here should inform planning for sustainable development of a high-growth aquaculture industry, critical to feeding an expanding global population[54,60] while balancing the imperative for healthy freshwater and marine environments, and the preservation of antimicrobial efficacy for future generations.

## Methods

**Literature review**. We searched PubMed, Web of Science, Scopus, the China National Knowledge Infrastructure database, and grey literature repositories (AGRIS, CGIAR FISH, IFPRI, WorldFish) for point prevalence surveys (PPS) of phenotypic antimicrobial resistance in bacteria isolated from aquatic animals for human consumption in Asia. The search identified surveys published between January 1, 2000, and September 30, 2019, with samples originating from cultured or wild-caught aquatic animals or their products for human consumption. Reviews and meta-analyses were excluded, as were studies with samples originated from bivalve molluscs or ornamental fish (Supplementary Note 1; Supplementary Data 2). The literature search and systematic review were guided by the Preferred Reporting Items for Systematic reviews and Meta-Analyses (PRISMA) statement and research synthesis norms[61] (Supplementary Table S4).

**Data extraction**. Collected data included study details (record identifier, author, year of publication, country), sampling details (latitude and longitude of sample collection, sampling dates, animal species, and whether cultured or wild-caught, number and origin of samples [skin; tissue; intestinal contents; or lesion] collected, health status of animal, history of antimicrobial use), and susceptibility testing details (bacterial genera, species, and strain, number of isolates subjected to antimicrobial susceptibility testing, susceptibility testing method, breakpoints and guidelines used, drug class, compound, resistance rate) (Supplementary Note 2). Where data was missing or required clarifications, the corresponding author was contacted by email. A total of 44 emails were sent requesting clarification, and 15 responses were received (34% response rate). Records were excluded when no response could be provided by the authors, and unclear data precluded further analysis.

There were 104 unique species or groups of species represented in our dataset. To facilitate analysis, species were aggregated into six species groups (Supplementary Fig. S12) reflective of aquatic animal and type of aquatic environment: marine fish, freshwater fish, brackish water fish, shrimp, and a mixed group of aquatic animals spanning these groups and for which resistance rates could not be disaggregated. The remaining species were pooled into a sixth group that included other crustaceans (crab), cephalopods (squid), gastropod molluscs (abalone), amphibians (frogs and salamanders), echinoderms (sea cucumbers and sea urchins), and reptiles (turtles).

We analyzed antimicrobial resistance in surveys at the bacterial genera level. This level of taxonomy was completely available (no missing entries) in our database, whereas the more granular bacterial species and strain level data were either not consistently provided or could not be disaggregated (143 surveys; 19%).

**Temporal trends**. We used the percentage of antimicrobial compounds in each survey with resistance rates exceeding 50% (P50) as a summary metric of multi-drug resistance. The P50 metric was used in the analysis of temporal trends and for geospatial modeling. We compared P50 with two additional metrics: P30 (calculated as the percentage of antimicrobial compounds in each survey with resistance exceeding 30%) and mean resistance (calculated as the total number of resistant isolates divided by the number of isolates * the number of antibiotics tested in each survey). Across all surveys, there is a positive correlation between P50 and mean resistance (Pearson's correlation coefficient = 0.9596) (Supplementary Note 3; Supplementary Fig. S19). Temporal trends in the P50 for each survey from cultured and wild-caught aquatic food animals were analyzed using generalized linear model regressions with quasibinomial error distribution weighted by the log of the number of isolates in each survey subjected to susceptibility testing. Root mean square error (RMSE) was used to evaluate the goodness of fit for the temporal trends regression models. RMSE indicated model fits were moderate (RMSE-cultured = 0.223; and RMSEwild caught = 0.235), consistent with both the scattered nature and scarcity of the data. The 95% confidence intervals were generated as the fitted values +/− 1.96 * standard error of the fitted value.

One-way analysis of variance (ANOVA) tests conducted on arcsine transformed P50 values were used to analyze the significance of the difference in mean P50 between survey characteristic groups, including culture or wild-caught, history of antimicrobial use, and health status of aquatic animals sampled (Supplementary Figs. S7, S8, and S10). Fisher's exact test was used to compare prevalence between two-time points.

**AMR in foodborne pathogens**. We calculated the pooled prevalence of resistance from individual pathogen-drug resistance rates to report resistance in foodborne pathogens. Resistance in foodborne pathogens of aquatic animal origin (*Vibrio* spp., *Streptococcus* spp. and *Aeromonas* spp.) was guided by antimicrobial compounds of relevance for therapeutic use in human clinical settings. We also analyzed resistance in *E.coli* as a marker of potential human and terrestrial animal influence. Resistance rates were calculated for these pathogens from samples originating from marine fish, freshwater fish, and shrimp groups using The Clinical and Laboratory Standards Institute (CLSI M45 and M100) and WHO Advisory Group on Integrated Surveillance of Antimicrobial Resistance (AGISAR)[62] pathogen-drug susceptibility testing guidelines (Supplementary Table S1). The 95% confidence interval was calculated on the resistance proportions.

**Geospatial modeling**. We mapped antimicrobial resistance in freshwater and marine environments at a resolution of 0.08333 decimal degrees, or approximately 10 km at the equator, by interpolating P50 values between point prevalence surveys. In freshwater environments, a two-step procedure was used, in which we first trained multiple child models, and subsequently stacked predictions from these models for universal kriging. Stacked generalization ensemble approaches have been used to model population-level health metrics[63], and previous work[64] has demonstrated that stacking models improve prediction accuracy compared with individual predictive models. The two-step procedure captures both the relationship between P50 and environmental and anthropogenic covariates, as well as spatial autocorrelation in the distribution of P50. This approach has recently been used to model the distribution of AMR in terrestrial animals in low-income and middle-income countries[26]. Although there is an inevitable trade-off in improved accuracy at the expense of reduced model interpretability, we chose an ensemble approach for prediction accuracy as the focus of our study was to produce the best possible maps of AMR rather than risk factor identification.

In total, 500 surveys were used to map AMR in freshwater environments. Where surveys did not provide precise sampling coordinates, we assigned coordinates at random within a geographic uncertainty range associated with the given sampling location. The uncertainty range was calculated as the mean of the distance in kilometers in the X and Y directions from the centroid to the boundaries of the smallest available administrative unit or place name provided in the survey. We used a random binarization procedure to transform the P50 values into presence (P50 = 1) and absence (P50 = 0) of resistance. We then generated and distributed pseudo-absence points to provide additional covariate values not associated with presence (P50 = 0) using stratified random sampling proportional to the human population density to account for potential P50 observation bias in more densely populated areas. Pseudo-absence points were sampled within a geographic radius of 20 to 500 km from presence points (Supplementary Note 4).

In the first step, environmental and anthropogenic covariates relevant to the freshwater environment were used to train three child models to quantify the association between P50 and these covariates. The 13 covariates were: accessibility to cities; gross domestic product; irrigated land percentage; minimum monthly temperature; terrestrial livestock P50; terrestrial livestock antimicrobial use; human population density; and population densities of cattle, pigs raised intensively, pigs raised semi-intensively, pigs raised extensively, chickens raised intensively, and

chickens raised extensively (Supplementary Fig. S13 and Table S2). Child models included boosted regression trees (BRT)[65]; least absolute shrinkage and selection operator applied to logistic regression (LASSO-GLM)[66]; and overlapped grouped LASSO penalties for General Additive Models selection (LASSO-GAM)[67] (Supplementary Fig. S14). BRT models have demonstrated good predictive performance in handling non-linear relationships and interactions amongst a diverse set of covariates and have been frequently used to model disease distribution[25,65,68,69]. By generating and combining a collection of models (decision trees) in a sequential stepwise fashion, boosting reduces both bias and variance while protecting against model overfitting. In addition, BRT models are insensitive to outliers[65]. LASSO regression models—here applied to GLM and GAM—facilitate efficient covariate selection by shrinking some regression coefficients and setting others with minor contributions to zero. These features enable a robust selection of covariates, reducing model complexity and strengthening predictive performance. Child models were fitted using three-fold spatial cross-validation aligned to the Asia sub-regions in our study (Supplementary Fig. S3). This cross-validation procedure takes observations from the training and validation sets which are geographically independent, guarding against overfitting and selection of models with poor capacity to predict to new areas[68]. Models were bootstrapped 10 times to account for variability introduced in the geographic assignment, the random binarization of P50 values, and the stratified random sampling of pseudo-absence points. The mean value of the area under the receiver operator characteristic curve (AUC) for all bootstraps was used to evaluate model predictive ability.

In the second step, predictions from child models were stacked and used as covariates for universal kriging. We fit a Matern variogram at a maximum distance of 500 km, which is where the semi-variogram attained the range. The kriging procedure was weighted by the number of isolates at each location. The output of the kriging procedure was a map of predicted freshwater resistance levels, as well as a map of kriging variance quantifying the spatial interpolation uncertainty (Supplementary Fig. S17). We also produced a "need for surveillance" index map for use in identifying optimal locations for future surveys, calculated as the kriging variance (uncertainty) weighted by the product of human population density[70] and inland aquaculture production volume[2] (described below). We further quantified uncertainty in the interpolation of P50 values by generating a map of the 95% confidence interval on the predicted P50 values (Supplementary Note 4; Supplementary Fig. S18).

In marine environments, a root mean square error (RMSE)-weighted ensemble model was used to map AMR. In contrast to the freshwater model, no association between marine covariates and P50 was identified, and we interpolate P50 values using survey coordinates in the marine AMR map. Surveys from inland freshwater sites were excluded. Wild-caught marine animals sampled at land-based post-harvest sites were randomly assigned coordinates to open ocean within a radius of .54 to 81 nautical miles (1 to 150 km) from their nearest coastal location (Supplementary Fig. S15). The marine data set consisted of two groups of surveys: (i) surveys from animals sampled at land-based post-harvest sites randomly assigned to open ocean; and (ii) surveys originating from marine, coastal marine, brackish water, and coastal brackish water sampling locations (Supplementary Note 4).

This marine data set (*n* = 322) was used to produce inverse distance weighted, natural neighbor, and ordinary kriging models. These models were subsequently stacked and weighted according to their root mean square error to capture the fit and variance of each model in the final ensemble model. The weights were taken as the inverse of the RMSE of each constituent model divided by the sum of RMSE for all models and expressed as their relative proportion in the final RMSE-weighted marine AMR ensemble model (Supplementary Table S3). A transparency function was added proportionally to the spatial kernel density of surveys at a bandwidth of 8.333 decimal degrees to reflect the density of the geographic distribution of surveys contributing to the final marine P50 map.

**Optimizing locations for future surveillance**. We used the "need for surveillance" index map for freshwater AMR to identify locations for 50 hypothetical surveys—the rounded mean number of annual surveys between 2010 and 2019—that could be conducted across Asia next year. The "need for surveillance" index was calculated as the product of the uncertainty from the spatial interpolation, human population density[70], and inland aquaculture production volume[2]. Human population and aquaculture production terms were standardized to range between 0 and 1 to give equal importance in determining the locations of future surveys. This function weights the necessity for surveillance to locations where the potential exposure risk and impact of AMR is greatest on the aquaculture industry and human health—via local consumption[71] and the cyclical exchange of resistant bacteria and their determinants across humans, aquaculture, and the aquatic environment.

We followed an approach proposed by Zhao et al.[72] that exploits a key feature associated with each additional survey conducted, reducing the uncertainty of the geospatial model in its surrounding area. The survey locations were optimized to reduce uncertainty as quantified through the "need for surveillance" index, thereby maximizing information gained for each successive survey (Supplementary Note 4).

Data analysis was conducted in R version 3.6.3 (Supplementary Software).

**Reporting summary**. Further information on research design is available in the Nature Research Reporting Summary linked to this article.

## Data availability

All data generated from this study are available on the Zenodo public repository: https://doi.org/10.5281/zenodo.4615703[73] and at https://resistancebank.org.

## Code availability

R code used to analyze the data and generate the models are available on the Zenodo public repository: https://doi.org/10.5281/zenodo.4615703[73].

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

## Acknowledgements

T.P.V.B., C.Z., and Y.W. were supported by the Swiss National Science Foundation and the Branco Weiss Foundation.

## Author contributions

D.S., M.G. and T.P.V.B. designed the research; D.S., C.Z. and Y.W. collected the data; D.S., C.Z., D.G.J.L., M.G. and T.P.V.B. analyzed data; D.S., C.Z., D.G.J.L., M.G. and T.P.V.B. wrote the paper.

## Competing interests

The authors declare no competing interests.
