## [Peer Review File · Nature Communications]

Twenty-year trends in antimicrobial resistance from aquaculture and fisheries in AsiaREVIEWER COMMENTS

Reviewer #1 (Remarks to the Author):

Review of:

Twenty-year trends in antimicrobial resistance from aquaculture and fisheries in Asia

This is a methodological study using data extracted from a systematic literature review. This covers an interesting area and the authors used epidemiological measures, spatial statistics and machine learning analytics to assess data extracted from their systematic review to show twenty-year trends in antimicrobial resistance from aquaculture and fisheries in Asia.

However, the review materials were not made available which hinder the ability to perform a critical peer review. Thus, as a minimum, all the study data must be provided before any publication of the study. Additionally, in my opinion the methodological approach used, the reporting of the study analytical framework and conclusion of findings requires a large number of improvements.

The authors did a systematic literature review but there is no reported critical assessment of studies or data quality. Further, doing this review, the authors excluded other reviews and meta-analyses. This could lead into omitting literature and data that the study search strategy could not identify!

Major concerns:

The authors choose to combine data at the genera level. However, no information regarding how this might have influenced any findings are given.

The authors base their entire study on the P50 criteria. Thus, basically any occurrence of resistance in classes or bacteria at 49% and below will be equal to zero and anything from 50 and above 1. If the information in Figure 2 is correct, then this would remove all variation of resistance among *Streptococcus* and information from most drug classes except in *E. coli* from South Asia and *Aeromonas* in Western and Southern Asia. No validation of this approach is given.

Without access to the exact data it is very difficult to evaluate whether the 10 miles distance has any meaning or the trends can only be interpreted at a country/regional level

Introduction:

Line 76: Considering the amount of time and studies where people (actually including myself) have tried to establish links between fish and humans, wouldn't it also be an option to write that there likely in many cases is limited transmission, except of course the known issues with *Vibrio vulnificus* and *Streptococcus agalactiae*.

Materials and methods

Lines 371-373, the authors excluded some records, yet the data in these records could have been assessed in sensitivity analyses. The authors did not provide a list of excluded records which should be reported - may be in the appendix.

Lines 399-400, the authors reported that "Resistance rates were excluded from analysis if fewer than 10 isolates were subjected to susceptibility testing for a specific pathogen-drug combination in each species group.". The authors should report the number of these surveys along with volume and type of excluded data (isolates).

Line 385: The use of P50 is a very critical step in the entire study. However, no argument for using this is provided and seemingly no concerns regarding whether this value with a limited number of isolates could be very uncertain. Also with absolutely no data provided it is impossible for the potential critical reader to verify this. Why do they not use actual values?

Line 391-392: There can be major differences in resistance between the different species of these

genera.

Line 400: calculate 95% confidence intervals, yes, but if they anyway just use P50 then this seems redundant.

Line 408 – geospatial modelling: I would like to question whether this is possible and how the authors have verified that this is a valid approach. To perform a geospatial mapping the authors need data on the exact origin of the data. Where did they get this from? Without access to the background data this is not possible to evaluate. Assigning geolocation at random within a country will include a major uncertainty and even for small countries two different surveys could be many miles apart. My best estimate would be that any resolution below the country level would be taking the data too far.

Line 462-473: This seems like an academic exercise. The authors include human density and inland aquaculture production in a model to suggest sites for additional surveys. However, they provide no argument for including this with equal weights or even for example why the human density should even be included. I can only speculate that the assumption is either that humans might contaminate the aquaculture production or that they assume that most aquaculture production are consumed locally. However, the latter would depend on type of production, since at least some are widely transported and even exported. It also seems from the appendix that they actually uses additional information, but exact how is difficult to see.

Results:

Line 127-153: As far as I can judge the authors do not take into account that resistance to some drugs on average is lower than to others, but simply stick to the P50 across all classes. This seems very strange since it is well-known that resistance is more easily acquired to some drugs than to others.

I also noted some very high levels of resistance to colistin. While it naturally can not be excluded that this is truly acquired resistance, it should be remembered that intrinsic colistin resistance is quite common and can greatly influence any conclusions, especially when as the authors grouping is done at the genera level. Also +50% carbapenem resistance. This is higher than in many clinical sites.

Other comments

This study recycled a large amount of text from another study (Zhao et al.) that was submitted along with this manuscript. The authors should revise the two manuscripts to assure that the two papers are properly rephrased.

Another flaw in this paper is poor citation for reported methods. The employed methods have already been in use in similar literature. Therefore, please, cite properly the details under the methods section in the manuscript and the appendix.

It is obvious that potential inclusion of survey new data from additional sources could influence the study findings. There could be a significant information gain here that might change the data interpretation and conclusions. This aspect was only briefly covered in the manuscript under the discussion section.

In line with that, the authors did not meticulously discuss lack of adequate survey coverage and the influence of underreporting on their findings. For instance, in lines 278-281, authors stated "Low AMR in the peri-urban environment of Guangzhou in southern China was unexpected considering the comparatively dense human population, prevalence of integrated livestock-aquaculture farming systems and broad agricultural and human connectivity". These findings could be attributed into biased reporting in literature which merit a critical assessment under the discussion section.

Appendix

In the appendix, line 54, the authors stated "Search strategies were tailored to individual databases) but they didn't provide such strategies - the authors provided only the full search query for PubMed. Details on the search strategy is missing for grey literature - it is not clear if the authors have searched all titles in the sources, or done a hand search of references ,etc. For transparency of reporting and for performing a critical review, the authors had to provide this information.

In the appendix, the authors described in good details what they have done but the motivation for their decisions was not properly covered. Further, in some instances, reporting of model building process was little confusing. For example in the appendix under the marine protocol sub-section, line 264, the authors stated "an averaged ensemble model was used ...", without reporting the rationale for using this approach - for example " weighted averaged models" might be a more appropriate choice here as not all the models have equal importance.

In reference to employed ensemble models, the authors should critically discuss the models' complexity in relation to generalization error of bias and variance. Authors did not discuss the main limitations of their models in relation to reduction of interpretability of complex models, as well as bias and noise considering the source of data.

Considering reported data in illustrations, some figures and maps lacks proper data elements. For instance, figure 1 in the manuscript could be augmented with information on reported number of surveys and/or isolates per year. A proper color legend should be provided here too.

References that might be used to improve reporting of this study findings

- Wei, D., Tang, K., Wang, Q., Estill, J., Yao, L., Wang, X., Chen, Y. and Yang, K. (2016), The use of GRADE approach in systematic reviews of animal studies. *Journal of Evidence-Based Medicine*, 9: 98-104. <https://doi.org/10.1111/jebm.12198>
- Percie du Sert N, Hurst V, Ahluwalia A, Alam S, Avey MT, et al. (2020) The ARRIVE guidelines 2.0: Updated guidelines for reporting animal research. *PLOS Biology* 18(7): e3000410. <https://doi.org/10.1371/journal.pbio.3000410>

Reviewer #2 (Remarks to the Author):

This study by Schar et al. sets out to investigate the temporal trends in antimicrobial resistance (AMR) in aquatic animal products in Asia, and it relies on reported data and geospatial modeling. AMR in the environment is a major threat to human health at global scale, and aquaculture is one of the major drivers, which has been widely studied, but large scale analysis has seldom been done so far. Therefore, it is a timely study, and will provide a bench mark for further surveillance of environmental AMR and help guide policy to mitigate the spread of AMR in animal industry. Overall, I think about 70% of the paper is more towards a meta-analysis, and with about 30% effort in modeling and prediction, which represents the major novelty aspect of the paper. However, I have some major concerns that might need to be addressed before this can be accepted in a board appealing journal-Nature Communications. Let me be specific as below.

1. I think, by using satellite data, one can map out aquaculture facilities/areas in the region, then look at the distribution of the 749-point prevalence surveys on top of the map derived from geospatial information. Further data on volume of production maybe available in governmental statistics year book? It will also allow to analyze the temporal trends in areas and (production perhaps). This information is important is postulating future surveillance foci considering both point data source but also weighted against area/production scale;
2. From the SI information, I can see that in general, among the 749 surveys, more surveys were conducted in more recent years. Should we also discuss the this potential bias? E.g. in 2002
3. Further to my point 2, is there any assessment regarding the temporal-spatial matching? I

guess this information will be useful in discussing the bias and uncertainty in modeling and also general conclusion. Does the coverage of regions correspond to the proportion of the production volume of that region? How does it matter in terms of modeling? I bet it does, particularly in directing future surveillance efforts;

4. In figure s8, the information as to antimicrobial use prior to sampling, this is a bit vague and not so robust, how long ago?

5. In figure s9, the correlation between sanitation and prevalence of AMR of wild caught samples, are these sample from freshwater or marine?

In summary, I think this study is of value, and will be an important literature for future studies and surveillance, particularly the compilation of existing survey data in one paper. As I have said earlier, the major strength is geospatial modeling, but I am not a modeler, so I suggest the final decision should weigh towards those reviewers with modeling expertise. In my opinion, if the modeling is robust, the paper can be potential accepted, though it needs much additional data. Also, it may, during the revision, seek collaborator from major countries, as local knowledge of the geography, production and AMR research related to topic (possibly with inclusion of some publications in national language) will increase the robustness of the overall analysis.

Reviewer #3 (Remarks to the Author):

This is a nicely written and interesting paper exploring antimicrobial resistance. I have focused my comments on the statistical analyses, which are generally described well.

1. For the linear regression used for temporal trends, what model goodness of fit measures were examined to ensure adequate fit?

2. Lines 403-406: Please comment on how you checked the assumptions underlying an ANOVA, and whether these assumptions were met.

3. You might like to mention as a limitation/scope for future work to consider incorporating currents into the analysis, especially given the apparent impact of human/cattle antimicrobial consumption and water contamination.

Supplementary information

4. Line 217: Is it possible to elaborate on/provide a reference for replicating the freshwater dataset 5 times?

5. Lines 219-221: Please mention the number of surveys with precise sampling coordinates provided.

Minor

6. Line 307: Suggest not italicising "overlap area" and "neighborhood area"

7. For the legends on Figures 3 and 5, suggest changing P50% to P50.

8. Figure S16: Were there no positive associations, or are they simply not visible due to tight ranges?

REVIEWER COMMENTS

Reviewer #1 (Remarks to the Author):

Response to Reviewer #1

We thank the reviewer for the thoughtful and detailed review.

Review of:

Twenty-year trends in antimicrobial resistance from aquaculture and fisheries in Asia

This is a methodological study using data extracted from a systematic literature review. This covers an interesting area and the authors used epidemiological measures, spatial statistics and machine learning analytics to assess data extracted from their systematic review to show twenty-

year trends in antimicrobial resistance from aquaculture and fisheries in Asia.

However, the review materials were not made available which hinder the ability to perform a critical peer review. Thus, as a minimum, all the study data must be provided before any publication of the study. Additionally, in my opinion the methodological approach used, the reporting of the study analytical framework and conclusion of findings requires a large number of improvements.

R1.1 Thank you. We have provided the raw data set as supplementary data 1.

The authors did a systematic literature review but there is no reported critical assessment of studies or data quality. Further, doing this review, the authors excluded other reviews and meta-analyses. This could lead into omitting literature and data that the study search strategy could not identify!

R1.2 Thank you. Our pre-defined eligibility criteria (SI L33-35) included point prevalence surveys of antimicrobial resistance in bacterial pathogens of production significance from cultured or wild caught aquatic animals or their products conducted in Asia between 2000 and 2019. Reviews and meta-analyses do not report full primary source data required for our study and were excluded (supplementary data 2). We have now provided the list of excluded records as supplementary data 2.

We assessed data quality in our search, excluding records where no source or methodology for derivation of data was provided; resistance rates were unclear or missing; no geographic information on survey location was provided; samples originated from imported products; or samples were not clearly identified as originating from an aquatic animal or animal product. We add this context (SI L45-48) and discuss data quality and the challenges inherent in standardizing data across a large volume ($n=749$) of point prevalence surveys (L330-333). We also discuss limitations associated with geographic variability in the surveys, which could conceivably introduce bias (L359-366).

Major concerns:

The authors choose to combine data at the genera level. However, no information regarding how this might have influenced any findings are given.

R1.3 We thank the reviewer for highlighting this point. We conducted a sensitivity analysis, re-analyzing the data at the species and strain level – the most granular level of data extracted in our database. The consistency in the significance level and directionality of the P50 temporal trends regressions, and in goodness of fit (cultured: $RMSE_{\text{genera}} = 0.223$, $RMSE_{\text{species}} = 0.226$; wild caught: $RMSE_{\text{genera}} = 0.235$, $RMSE_{\text{species}} = 0.221$) suggest that taxonomy classification does not influence the central findings of our initial analysis: that P50 in cultured aquatic animals plateaus at 33% and in decreases in wild caught animals.

Annual trends in the proportion of drugs with resistance greater than 50% (P50) in each survey by bacterial taxonomy. Top row are surveys analyzed at genera level; bottom row are surveys analyzed at species or strain level. **(a,b)** P50 for cultured aquatic animals; and **(c,d)** wild caught aquatic animals. Goodness of fit is assessed using root mean square error (RMSE). A solid line indicates statistical significance ($p < 0.05$); 95% confidence intervals are shown in shaded areas.

Event-based surveillance surveys are inherently conducted under variable conditions and reporting structures, requiring pooling in order to build critical sample size and make statistical inferences. We chose to pool at the bacterial genera level because this level was completely available (no missing entries) in our database, whereas the more granular species and strain level data were either not consistently provided or reported as multiple bacterial species and strains together where resistance rates could not be disaggregated (143 surveys; 19%). A description of our handling of bacterial genera, species and strain reporting is provided in our database legend (doi: 10.5281/zenodo.4609884).

Notably, in 2019, the U.S. FDA National Antimicrobial Resistance Monitoring System (NARMS) [<https://www.hhs.gov/sites/default/files/7.3-mcdermott-narms-508.pdf>] initiated a pilot surveillance program for pathogens from seafood. This pilot study design targets *Vibrio*, *Aeromonas* and *Enterococcus* spp. reported at the genera level.

We acknowledge that routine surveillance, consistently reported at the bacterial species level may be important in identifying trends of local or context-specific significance (e.g. *Vibrio parahaemolyticus* in coastal brackish water aquaculture systems) – however the aim of our study was rather to draw a first overview of the trends over twenty years in Asia across a broad range of bacteria. We reflect this point in the discussion (L226-228):

“these trends may not capture more granular country-level resistance profiles in select pathogen species from specific aquatic animal production settings (Supplementary Fig. S11).”

We also add this context to the methods (L422-425) and supplementary methods (SI L309-312), and thank the reviewer for helping us most effectively communicate our approach and findings.

“We analyzed antimicrobial resistance in surveys at the bacterial genera level. This level of taxonomy was completely available (no missing entries) in our database, whereas the more granular bacterial species and strain level data were either not consistently provided or could not be disaggregated (143 surveys; 19%).”

The authors base their entire study on the P50 criteria. Thus, basically any occurrence of resistance in classes or bacteria at 49% and below will be equal to zero and anything from 50 and above 1. If the information in Figure 2 is correct, then this would remove all variation of resistance among *Streptococcus* and information from most drug classes except in *E. coli* from South Asia and *Aeromonas* in Western and Southern Asia. No validation of this approach is given.

R1.4 We would like to provide an important clarification regarding interpretation of our findings. We use pooled prevalence of individual pathogen-drug resistance rates to report resistance in foodborne pathogens specifically (Figure 2), and the P50 summary metric of multi-drug resistance to analyze temporal trends across all surveys (Figure 1) and for geospatial modeling. The P50 metric is not comprised of the pooled prevalence of resistance illustrated in figure 2, but rather is calculated for each survey as the percentage of antimicrobial compounds in each survey with resistance exceeding 50%. The distribution of survey P50 values by year is represented in Figure 1.

We have added language in the results (L134-135) and restructured subsection headings and language in the methods (L427-430; L451-453) to clarify that the P50 metric is used in the temporal trends analysis and geospatial modeling and is calculated directly from each individual survey, while pooled prevalence of individual resistance rates is used in the reporting of AMR in foodborne pathogens specifically.

We also compare P50 with two additional metrics: P30 (calculated as the percentage of antimicrobial compounds in each survey with resistance exceeding 30%) and mean resistance, calculated as the total number of resistant isolates divided by the number of isolates * number of antibiotics tested in each survey (supplementary data 1).

Fig. S19. Annual trends in multi-drug resistance. Surveys from cultured aquatic animals (top row, $n = 558$); surveys from wild caught aquatic animals (bottom row, $n = 81$). **(a,b)** the proportion of drugs with resistance greater than 50% (P50) in each survey; **(c,d)** mean resistance calculated as resistant isolates divided by the number of isolates * number of antibiotics tested in each survey; and **(e,f)** the proportion of drugs with resistance greater than 30% (P30) in each survey. Regression lines are fit using generalized linear models, with a solid line indicating statistical significance ($p < 0.05$); 95% confidence intervals are shown in shaded areas. Goodness of fit is assessed using root mean square error (RMSE).

There is a positive correlation between P50 and mean resistance in our study (Pearson's correlation coefficient: all surveys = 0.9596; cultured = 0.965; wild caught = 0.937).

In cultured animals, P50 (RMSE = 0.223 ; coefficient = 0.004; $p = 0.633$) and Mean (RMSE = 0.207 ; coefficient = 0.002 ; $p = 0.758$) were comparable with a positive coefficient slope, whereas for P30 (RMSE = 0.235 ; coefficient = -0.012; $p = 0.158$) the slope is negative. In wild caught animals, the Mean model fit improved (RMSE = 0.176; coefficient = -0.065; $p = 0.002$) when compared with P50 (RMSE = 0.235; coefficient = -0.085; $p = 0.003$) and P30 metrics (RMSE = 0.262; coefficient = -0.106; $p = 0.001$), however the directionality of the trends is consistent and remain statistically significant across all metrics.

The results of this comparison of metrics shows that there is consistency in the results obtained with different metrics. We add the temporal trends comparing P50, P30, and mean resistance as Supplementary Figure S19.

As a multi-drug resistance metric, P50 is valuable in quantifying the level of resistance displayed by a particular bacteria across drugs in a survey. The P50 uses the 50% threshold of resistance as a marker of potential therapeutic efficacy – namely, the probability that antimicrobial chemotherapy options could be effective. Because our systematic review inclusion criteria

restricted survey eligibility to bacterial pathogens of production significance or aquatic animal-associated bacterial zoonoses (SI L33-34)—and 513/749 (68.5%) surveys involved recognized bacterial zoonoses (*Vibrio*, *Aeromonas*, *Streptococcus*, *Edwardsiella*, and *E.coli*)—we assessed that the P50 metric was better suited to communicate our findings in the context of clinical therapeutic efficacy.

We incorporate this into the methods (L430-435):

“We compared P50 with two additional metrics: P30 (calculated as the percentage of antimicrobial compounds in each survey with resistance exceeding 30%) and mean resistance (calculated as the total number of resistant isolates divided by the number of isolates * the number of antibiotics tested in each survey). Across all surveys, there is a positive correlation between P50 and mean resistance (Pearson’s correlation coefficient = 0.9596) (Supplementary Fig. S19).”

And we detail the comparison of metrics in the supplementary information (SI L272-287) as well as adding the comparison plots as Supplementary Fig. S19:

“In cultured animals, P50 (RMSE =0.223 ; coefficient = 0.004; p = 0.633) and Mean (RMSE =0.207 ; coefficient = 0.002 ; p = 0.758) were comparable with a positive coefficient slope, whereas for P30 (RMSE =0.235 ; coefficient = -0.012; p = 0.158) the slope is negative. In wild caught animals, the Mean model fit improved (RMSE = 0.176; coefficient = -0.065; p = 0.002) when compared with P50 (RMSE = 0.235; coefficient = -0.085; p = 0.003) and P30 metrics (RMSE = 0.262; coefficient = -0.106; p = 0.001), however the directionality of the trends is consistent and remain statistically significant across all metrics (Supplementary Fig. S19).

Using a 50% threshold of resistance, the P50 metric serves as a useful marker of potential therapeutic efficacy – namely, the probability that antimicrobial chemotherapy options could be effective. Across surveys in our study, 68.5% (n=513) involved recognized bacterial zoonoses. We therefore assessed that the P50 metric was better suited to communicate our findings in the context of clinical therapeutic efficacy.”

Without access to the exact data it is very difficult to evaluate whether the 10 miles distance has any meaning or the trends can only be interpreted at a country/regional level

R1.5 The raw data set has been provided as supplementary data 1. Our approach to geospatial analysis and the 10-km resolution of our AMR maps is articulated in R1.12 below.

Introduction:

Line 76: Considering the amount of time and studies where people (actually including myself) have tried to establish links between fish and humans, wouldn't it also be an option to write that there likely in many cases is limited transmission, except of cause the know issues with *Vibrio vulnificus* and *Streptococcus agalactiae*.

R1.6 We have framed this section of the introduction to highlight the unique characteristics of aquatic food animal supply chains that present the potential for dissemination of AMR. We acknowledge that transmission is dynamic and variable by context, and that current evidence—while growing—is not definitive. We have added this context accordingly (L76-79):

“In aquaculture and fisheries—although more limited than in terrestrial animals—the evidence concerning antimicrobial resistance has also expanded over the last decade^{10,11,12,13}. Documenting the movement of resistance determinants in aquatic settings presents challenges, and the body of evidence from which to draw conclusions on antimicrobial resistance transference between aquatic animals and humans remains very limited.”

Importantly, we expect our work will serve as a catalyst for further epidemiological investigation that will help expand the understanding of antimicrobial resistance transference.

Materials and methods

Lines 371-373, the authors excluded some records, yet the data in these records could have been assessed in sensitivity analyses. The authors did not provide a list of excluded records which should be reported - may be in the appendix.

R1.7 We have now provided a list of all included and excluded records (supplementary data 2). L371–373 (now L409-411) refer to records excluded where essential data fields were missing or the presentation of data did not permit extraction (e.g. no geographic survey location was provided; resistance rates were aggregated, representing multiple bacterial genera; or phenotypic AST was alluded to but not provided). In each case, attempts were made to contact the authors (n=44). In such cases where no response was received (n=29; 66%) or clarification could not be provided, the records were excluded. These records for which data could not be extracted are in supplementary data 2.

Lines 399-400, the authors reported that “Resistance rates were excluded from analysis if fewer than 10 isolates were subjected to susceptibility testing for a specific pathogen-drug combination in each species group.”. The authors should report the number of these surveys along with volume and type of excluded data (isolates).

R1.8 We would like to clarify that the reference (L399-400) to exclusion of resistance rates pertains only to the display in Figure 2 of pathogen-drug combinations for the four foodborne pathogens—a decision made to accommodate the large volume of pathogen-drug pairings in a graphically legible format in Figure 2. The narrative reporting of the pooled prevalence of resistance for these foodborne pathogens in the results (L134-157) does not exclude any isolates from the analysis. We have restructured the methods, adding a sub-heading “**AMR in foodborne pathogens**” (L451) to highlight the distinction in analyzing these data, and removing the reference to exclusion of isolates both in the main manuscript and the supplementary information. We maintain the reference in the figure legend to the Figure 2 display of resistance for pathogen-drug combinations with 10 or more isolates tested (L750-752). We are greatly appreciative to the reviewer for helping us to report our methods with greater precision.

Line 385: The use of P50 is a very critical step in the entire study. However, no argument for using this is provided and seemingly no concerns regarding whether this value with a limited number of isolates could be very uncertain. Also with absolutely no data provided it is impossible for the potential critical reader to verify this. Why do they not use actual values?

R1.9 We appreciate the reviewer's comment and the opportunity to strengthen the communication of our methodology.

We agree that surveys with a limited number of isolates may be associated with uncertainty. This uncertainty, however, is not specific to P50, but is inherent in any metric that summarizes resistance rates in order to provide an overall picture of multi-drug resistance and AMR trends.

To reflect these uncertainties, we have assigned statistical weight in both the temporal trends regressions and the geospatial analyses proportional to the number of isolates in each survey subjected to susceptibility testing.

We visually represent these uncertainties by displaying the distribution of P50 values in each year in Figure 1 and by adding a new map of the 95% confidence interval on the predicted values (Supplementary Fig. S18).

In reference to "actual values," we assume the reviewer is referencing individual resistance rates. While we do report the pooled prevalence from these individual resistance rates in foodborne pathogens (L134-157; Figure 2), the sample sizes for individual pathogen-drug combinations are currently insufficient to make maps based on robust statistical inference.

Multi-drug resistance indices help in summarizing the often complex, multi-faceted information of drug resistance, and have been widely used to communicate burden of resistance trends as has been frequently noted in the literature (Laxminarayan R et al. doi: 10.1136/bmjopen-2011-000135; Hughes JS et al. doi: 10.1136/bmjopen-2016-012040; Krumpferman PH. doi: 10.1128/aem.46.1.165-170.1983).

We provide the results of a comparison of multi-drug resistance metrics and justification for our use of P50 in R1.4.

Line 391-392: There can be major differences in resistance between the different species of these genera.

R1.10 We agree with the reviewer. We have chosen to report pooled prevalence of resistance at the bacterial genera level in this study as this was the level at which complete data was available. Specifically, 143 of 749 total surveys (19%) did not report or could not be disaggregated below the genera level; and 72 of 469 (15%) surveys from these four foodborne pathogens specifically did not report resistance at the species level. Acknowledging the important point raised by the reviewer, we add this in the discussion of limitations (L333-337):

“Resistance across species within bacterial genera can vary markedly. Our analyses of drug resistance were conducted at the bacterial genera level—the most granular level at which complete data were available. Although analysis of trends at the bacterial species level may be valuable in context-specific settings, the aim of our study was rather to document temporal trends over twenty years in Asia across a broad range of bacteria.”

Line 400: calculate 95% confidence intervals, yes, but if they anyway just use P50 then this seems redundant.

R1.11 The 95% confidence intervals referenced in L400 (now L461) are calculated on the pooled prevalence of individual pathogen-drug resistance rates for foodborne pathogens, not on P50.

We have clarified in the revised manuscript through an added subsection heading in the methods (L451) to clearly identify the handling of pooled prevalence of resistance in foodborne pathogens as distinct from the P50 metric used for the temporal trends analysis and geospatial modeling. For the temporal trends analysis (Figure 1), we apply 95% confidence intervals to the generalized linear model regression, generated as the fitted values $\pm 1.96 * \text{standard error of the fitted value}$ (L441-443).

Line 408 – geospatial modelling: I would like to question whether this is possible and how the authors have verified that this is a valid approach. To perform a geospatial mapping the authors need data on the exact origin of the data. Where did they get this from? Without access to the background data this is not possible to evaluate. Assigning geolocation at random within a country will include a major uncertainty and even for small countries two different surveys could be many miles apart. My best estimate would be that any resolution below the country level would be taking the data too far.

R1.12 We agree with the reviewer that geospatial models carry uncertainty. However, in the analysis, geolocation was not conducted at random within a country, but rather within an uncertainty range associated with the name of the smallest available administrative unit or place name mentioned in the survey (L481-483). This approach addresses the issue of geolocation consistently across different countries.

Concretely, we used the following procedure (L479-524). We extracted precise latitude and longitude coordinates for each survey where a specific sampling location (e.g. named farm or market) was provided (43 freshwater surveys; 26 marine surveys). For freshwater surveys where exact coordinates were not provided, coordinates were assigned within a geographic uncertainty range associated with the name of the smallest available administrative unit (L481 – 483) mentioned as the sampling location in the survey. We used 10 draws from Monte Carlo simulations to capture the uncertainty between possible locations within that administrative unit.

We present a histogram of the geographic uncertainty ranges associated with each survey (n=749) in our study:

Out of all of the surveys in our study, 338 (45%) were within a geographic uncertainty range of 20km. A total of 63%, 71% and 84% of all surveys were within uncertainty ranges of 40, 60, and 100 km respectively.

For marine surveys, coordinates were either in open water or on land. Survey locations on land were handled as follows: (1) wild caught marine animals sampled at land based post-harvest sites were assigned coordinates to open ocean within a radius of .54 to 81 nautical miles (1 to 150 km) from their nearest coastal location (L529-532; Supplementary Fig. S15)— a range that captures the distribution of local water artisanal and industrial fishing fleets in Asia, and falls within the 200 nautical mile distance from coastline of exclusive economic zones for marine resource exploration (UN Convention on Law of the Sea, Article 57); and (2) coastal marine surveys with land based coordinates were redistributed to their nearest coastline. The final marine data set was comprised of these surveys and the surveys with extracted coordinates already in open water. A description of these protocols is provided in the supplementary information (SI L396) and the database legend (doi: 10.5281/zenodo.4609884).

Communicating geographical uncertainty is an important component in interpretation of our findings. Figure 4 shows a weighted uncertainty map associated with our AMR predictions in freshwater environments, and we have now added (1) a map of raw uncertainty (kriging variance); and (2) a map of 95% confidence interval on the predictions (Supplementary Figs. S17 and S18, respectively). We also present below the variogram plot showing spatial auto-correlation at 500km indicating that patterns of resistance are identifiable at sub-national level.

Fig. S18. 95% confidence interval on P50 predictions in freshwater environments.

Variogram plot from P50 in freshwater, displaying spatial auto-correlation at 500km.

We have also revised our marine map to produce a root mean square error (RMSE)-weighted ensemble model and add a supplementary table listing RMSE and weightings for each constituent model in the final ensemble (SI Table S3).

	RMSE	Model weights
Inverse distance weighted	0.2359776	31.7
Natural neighbor	0.2255336	33.2
Ordinary kriging	0.2131811	35.1

Table S3. Root mean square error (RMSE) weightings in the marine ensemble model. Weights are calculated as the inverse of the RMSE of each constituent model divided by the sum of RMSE for all models [$Weights = 1/(RMSE_i / \sum_{i=1}^3 RMSE_i)$], and expressed as their relative proportion.

In the absence of systematic surveillance, geospatial modeling approaches have been used to characterize the geographic distribution of many other diseases of global importance (malaria [Sinka ME et al. doi: 10.1186/1756-3305-5-69; Moyes CL et al. doi: 10.1186/s13071-016-1527-0], dengue [Bhatt S et al. doi: 10.1038/nature12060], lymphatic filariasis [Cano J et al. doi: 10.1186/s13071-014-0466-x], emerging zoonoses [Allen T et al. doi: 10.1038/s41467-017-00923-8], and ongoing efforts for AMR in humans), helping to prioritize action on a regional level.

We add language (L165-168) to enhance communication of our findings and assist the reader in understanding the uncertainty in our geospatial modeling.

“The interpolation of resistance in this study is associated with uncertainty. Variability in geolocation of surveys, in covariates, and in estimates of resistance contribute to this uncertainty, which is captured with a 95% confidence interval map on P50 predictions (Supplementary Fig. S18).”

Line 462-473: This seems like an academic exercise. The authors include human density and inland aquaculture production in a model to suggest sites for additional surveys. However, they provide no argument for including this with equal weights or even for example why the human density should even be included. I can only speculate that the assumption is either that humans might contaminate the aquaculture production or that they assume that most aquaculture production are consumed locally. However, the latter would depend on type of production, since at least some are widely transported and even exported. It also seems from the appendix that they actually uses additional information, but exact how is difficult to see.

R1.13 We thank the reviewer for encouraging a more thorough explanation of our methodology.

The Figure 4 map represents the need for surveillance, which is the product of three factors: 1) the level of uncertainty measured through kriging variance; 2) the level of exposure of the human population; and 3) the volume of inland aquaculture production. This approach prioritizes surveillance to those areas where the knowledge gained and potential impact of AMR on human health and aquaculture production is greatest. For example, high uncertainty in a remote, low human population density setting with low aquaculture production volume would assign a comparatively low need for surveillance under our approach. In such a setting, the level of AMR is effectively unknown (high uncertainty), but the potential exposure risk and impact of AMR on

human health and the aquaculture industry would be comparatively low and therefore such a location would not be a priority for surveillance.

We augment the protocol with additional detail in the supplementary information (SI L427-456) and have added language to the methods in the revised manuscript (L546-562):

“The “need for surveillance” index was calculated as the product of the uncertainty from the spatial interpolation, human population density⁶⁰, and inland aquaculture production volume². This function weights the necessity for surveillance to locations where the potential exposure risk and impact of AMR is greatest on the aquaculture industry and human health—via local consumption and the cyclical exchange of resistant bacteria and their determinants across humans, aquaculture, and the aquatic environment... The survey locations were determined to reduce uncertainty as quantified through the “need for surveillance” index, thereby maximizing information gained for each successive survey.”

We also refer to this in the discussion (L366-370). Our study responds to repeated calls in the literature and in policy fora for investments in targeted surveillance (Hay SI et al. doi: 10.1186/s12916-018-1073-z; O'Neill J. Tackling drug-resistant infections globally. [https://amr-review.org/sites/default/files/160525_Final%20paper_with%20cover.pdf]; Frost I et al. doi: 10.1093/jtm/taz036), and which are increasingly reflected in national AMR monitoring plans (U.S. NARMS 2021-2025) [<https://www.fda.gov/media/79976/download>]. We address these calls by applying an objective, data-driven approach to guide the identification of highest priority surveillance sites under competing needs for resource allocation. In particular, this approach may assist policy makers in developing a prioritization strategy for AMR surveillance by location when compared with surveillance schemes that distribute surveillance resources equally between administrative divisions.

Results:

Line 127-153: As far as I can judge the authors do not take into account that resistance to some drugs on average is lower than to others, but simply stick to the P50 across all classes. This seems very strange since it is well-known that resistance is more easily acquired to some drugs than to others.

R1.14 We have used pooled prevalence of individual pathogen-drug pairing resistance rates—not P50—to report the variation in resistance to drugs for foodborne pathogens. These are reported on L134-157 and illustrated in Figure 2.

I also noted some very high levels of resistance to colistin. While it naturally can not be excluded that this is truly acquired resistance, it should be remembered that intrinsic colistin resistance is quite common and can greatly influence any conclusions, especially when as the authors grouping is done at the genera level. Also +50% carbapenem resistance. This is higher than in many clinical sites.

R1.15 We thank the reviewer for this point regarding colistin resistance, which we have incorporated into the discussion (L269-277):

“Colistin resistance across a breadth of bacterial genera in Asia may reflect both intrinsic and acquired resistance mechanisms. Intrinsic resistance to polymyxins in Gram-negative bacteria has been documented⁴⁷. The possibility that a proportion of the colistin resistance identified is acquired cannot be ruled out, particularly through plasmid-mediated transfer amongst members of the Enterobacteriaceae family⁴⁸. Such resistance may be attributable to the prolific use of colistin and other polymyxins in the region, which is only recently evolving through national regulatory action, and the possible recruitment of *mcr*-family mobile resistance determinants, their broad dissemination through horizontal gene transfer and the wide distribution potential of aquatic environments^{14,48,49}.”

With respect to carbapenem resistance, we reviewed the EUCAST “Intrinsic resistance and unusual phenotypes v3.2 (February 2020)” [https://www.eucast.org/fileadmin/src/media/PDFs/EUCAST_files/Expert_Rules/2020/Intrinsic_Resistance_and_Unusual_Phenotypes_Tables_v3.2_20200225.pdf] tables and find no reference to intrinsic resistance to carbapenems in the pathogens in our study. We address elevated carbapenem resistance identified in *Aeromonas* in Western and Southern Asia specifically—as well as the significance of the carbapenemase encoding gene (*cphA*)—in the discussion (L239-250).

Other comments

This study recycled a large amount of text from another study (Zhao et al.) that was submitted along with this manuscript. The authors should revise the two manuscripts to assure that the two papers are properly rephrased.

R1.16 We have revised the manuscript accordingly (L546-562)

Another flaw in this paper is poor citation for reported methods. The employed methods have already been in use in similar literature. Therefore, please, cite properly the details under the methods section in the manuscript and the appendix.

R1.17 We have added references in both the methods and supplementary information, specifically:

Page, M. J. *et al.* The PRISMA 2020 statement: An updated guideline for reporting systematic reviews. *PLOS Med.* **18**, e1003583 (2021)
(L398 and SI L55-57)

Golding, N. *et al.* Mapping under-5 and neonatal mortality in Africa, 2000–15: a baseline analysis for the Sustainable Development Goals. *The Lancet* **390**, 2171–2182 (2017).
(L469 and SI L322)

Van Boeckel, T. P. *et al.* Global trends in antimicrobial resistance in animals in low- and middle-income countries. *Science* **365**, eaaw1944 (2019).
(L474 and SI L332)

Bhatt, S. *et al.* Improved prediction accuracy for disease risk mapping using Gaussian process stacked generalization. *J. R. Soc. Interface* **14**, 20170520 (2017).
(L469 and SI L334)

Elith, J., Leathwick, J. R. & Hastie, T. A working guide to boosted regression trees. *J. Anim. Ecol.* **77**, 802–813 (2008).
(L494; L498; L501 and SI L351; SI L371)

Tibshirani, R. Regression Shrinkage and Selection Via the Lasso. *J. R. Stat. Soc. Ser. B Methodol.* **58**, 267–288 (1996).
(L495 and SI L352)

Chouldechova, A. & Hastie, T. Generalized Additive Model Selection. *ArXiv150603850 Stat* (2015).
(L496 and SI L353)

Barbet-Massin, M., Jiguet, F., Albert, C. H. & Thuiller, W. Selecting pseudo-absences for species distribution models: how, where and how many?: How to use pseudo-absences in niche modelling? *Methods Ecol. Evol.* **3**, 327–338 (2012).
(SI L349)

Hijmans, R. J. Cross-validation of species distribution models: removing spatial sorting bias and calibration with a null model. *Ecology* **93**, 679–688 (2012).
(SI L360)

Gilbert, M. *et al.* Predicting the risk of avian influenza A H7N9 infection in live-poultry markets across Asia. *Nat. Commun.* **5**, 4116 (2014).
(L498; L508)

Additionally, the complete set of references for the environmental and anthropogenic covariates used to inform the freshwater models are provided in Table S2.

It is obvious that potential inclusion of survey new data from additional sources could influence the study findings. There could be a significant information gain here that might change the data interpretation and conclusions. This aspect was only briefly covered in the manuscript under the discussion section.

R1.18 Thank you. To the best of our knowledge, the multiple data sources assembled in our study constitute what is, to date, the most comprehensive database compiled from publicly accessible sources on antimicrobial resistance in bacteria from aquatic food animals in Asia.

Our dataset is based upon a comprehensive, structured systematic review of over 5,800 records across eight search engines, with over 1,900 originating from the China National Knowledge Infrastructure platform, an important element as China represents the largest share of aquaculture

production by volume globally. As with any search, there may be surveys that were not identified or available for review, and we have added this context to the discussion (L363-366):

“Despite our broad search parameters, surveys conducted in these countries and which were either not identified in our search or not available for review could potentially influence our findings. Similarly, in the absence of systematic surveillance, reliance on passive surveillance data presents variability in survey coverage which could introduce bias.”

We also underscore the value of future surveillance (L369-370):

“Such future surveillance will be essential to an enhanced and refined understanding of resistance trends in Asia.”

Finally, we highlight the current evidence base as comparatively thin, calling for systematic antimicrobial resistance surveillance in aquatic food animals moving forward (L377 - 378):

“A scale-up of an antimicrobial resistance surveillance architecture for aquatic food animals is urgently needed to fill gaps in AMR trends at national and sub-national levels.”

In line with that, the authors did not meticulously discuss lack of adequate survey coverage and the influence of underreporting on their findings. For instance, in lines 278-281, authors stated “Low AMR in the peri-urban environment of Guangzhou in southern China was unexpected considering the comparatively dense human population, prevalence of integrated livestock-aquaculture farming systems and broad agricultural and human connectivity” These findings could be attributed into biased reporting in literature which merit a critical assessment under the discussion section.

R1.19 We agree that geographic variability in survey coverage could introduce bias, as would be the case for any epidemiological assessment based on passive surveillance data. As articulated in R1.18, we have added this point to the discussion (L365-366).

We also provide a detailed discussion of the variability in geographic representation of surveys in our study, and the associated knowledge gaps in understanding AMR in aquatic food animals in Asia (L359-363; Supplementary Figs. S4 and S5). Importantly, through our optimization protocol, we aim to provide data-driven guidance to address the longstanding issue of variability in survey coverage. Specifically, we identify those areas that have, thus far, been poorly surveyed and have a high “need for surveillance” when viewed through the lens of potential impact on human health and the aquaculture industry. We then provide an evidence based prioritization for locations that would benefit most from future surveillance.

Appendix

In the appendix, line 54, the authors stated “Search strategies were tailored to individual databases) but they didn’t provide such strategies - the authors provided only the full search query for PubMed. Details on the search strategy is missing for grey literature – it is not clear if

the authors have searched all titles in the sources, or done a hand search of references ,etc. For transparency of reporting and for performing a critical review, the authors had to provide this information.

R1.20 Thank you for calling our attention to this omission. We have added the search strategies used for each database in the Supplementary Text (SI, L65 – 161):

PubMed

(Resistance OR "antibiotic resistance" OR "antimicrobial resistance") AND ("Escherichia coli" OR "E. coli" OR *vibrio* OR Photobacterium OR Aeromonas* OR Edwardsiella* OR Yersinia OR Pseudomonas* OR Flavobacter* OR Piscirickettsia OR Hepatobacter OR Francisella OR Chlamydia OR Mycobacter* OR Nocardia OR Streptococc* OR Lactococc* OR Aerococc* OR Renibacter* OR Clostridium OR Enterobacterium) AND (aquaculture OR aquatic OR *fish* OR shellfish OR marine OR freshwater OR carp OR catfish OR prawn OR salmon OR shrimp OR tilapia OR trout) AND (Asia OR "Southeast Asia" OR "South Asia" OR "East Asia" OR Mekong OR Afghanistan OR "American Samoa" OR Bahrain OR Bangladesh OR Bhutan OR "Brunei Darussalam" OR Cambodia OR China OR "Chinese Taipei" OR "Cook Islands" OR "Democratic People's Republic of Korea" OR Fiji OR "French Polynesia" OR Guam OR "Hong Kong" OR India OR Indonesia OR Iran OR Iraq OR Jordan OR Kiribati OR Korea OR Kuwait OR "Lao People's Democratic Republic" OR Lao OR Laos OR Lebanon OR Macau OR Malaysia OR Maldives OR "Marshall Islands" OR Micronesia OR Mongolia OR Myanmar OR Nauru OR Nepal OR "New Caledonia" OR Niue OR "Norfolk Island" OR "Northern Mariana Islands" OR Oman OR Pakistan OR Philippines OR Palau OR Palestine OR "Papua New Guinea" OR "Pitcairn Islands" OR Qatar OR Samoa OR "Saudi Arabia" OR Singapore OR "Solomon Islands" OR "Sri Lanka" OR "Syrian Arab Republic" OR Taiwan OR Thailand OR Timor-Leste OR Tokelau OR Tonga OR Turkey OR Tuvalu OR "United Arab Emirates" OR Vanuatu OR "Viet Nam" OR Vietnam OR "Wallis and Futuna Islands" OR Yemen)

Web of Science [All Databases, inclusive of Russian Science Citation Index and Korean Journal Database (KCI)]

TOPIC: (Resistance OR "antibiotic resistance" OR "antimicrobial resistance") AND TOPIC: ("Escherichia coli" OR "E. coli" OR *vibrio* OR Photobacterium OR Aeromonas* OR Edwardsiella* OR Yersinia OR Pseudomonas* OR Flavobacter* OR Piscirickettsia OR Hepatobacter OR Francisella OR Chlamydia OR Mycobacter* OR Nocardia OR Streptococc* OR Lactococc* OR Aerococc* OR Renibacter* OR Clostridium OR Enterobacterium) AND TOPIC: (aquaculture OR aquatic OR *fish* OR shellfish OR marine OR freshwater OR carp OR catfish OR prawn OR salmon OR shrimp OR tilapia OR trout) AND TOPIC: (Asia OR "Southeast Asia" OR "South Asia" OR "East Asia" OR Mekong OR Afghanistan OR "American Samoa" OR Bahrain OR Bangladesh OR Bhutan OR "Brunei Darussalam" OR Cambodia OR China OR "Chinese Taipei" OR "Cook Islands" OR "Democratic People's Republic of Korea" OR Fiji OR "French Polynesia" OR Guam OR "Hong Kong" OR India OR Indonesia OR Iran OR Iraq OR Jordan OR Kiribati OR Korea OR Kuwait OR "Lao People's Democratic Republic" OR Lao OR Laos OR Lebanon OR Macau OR Malaysia OR Maldives OR "Marshall Islands" OR Micronesia OR Mongolia OR Myanmar OR Nauru OR Nepal OR "New Caledonia" OR

Niue OR "Norfolk Island" OR "Northern Mariana Islands" OR Oman OR Pakistan OR Philippines OR Palau OR Palestine OR "Papua New Guinea" OR "Pitcairn Islands" OR Qatar OR Samoa OR "Saudi Arabia" OR Singapore OR "Solomon Islands" OR "Sri Lanka" OR "Syrian Arab Republic" OR Taiwan OR Thailand OR Timor-Leste OR Tokelau OR Tonga OR Turkey OR Tuvalu OR "United Arab Emirates" OR Vanuatu OR "Viet Nam" OR Vietnam OR "Wallis and Futuna Islands" OR Yemen)

Scopus

TITLE-ABS-KEY (resistance OR "antibiotic resistance" OR "antimicrobial resistance") AND TITLE-ABS-KEY ("Escherichia coli" OR "E. coli" OR *vibrio* OR photobacterium OR aeromonas* OR edwardsiell* OR yersinia OR pseudomonas* OR flavobacter* OR piscirickettsia OR hepatobacter OR francisella OR chlamydia OR mycobacter* OR nocardia OR streptococc* OR lactococc* OR aerococc* OR renibacter* OR clostridium OR enterobacterium) AND TITLE-ABS-KEY (aquaculture OR aquatic OR *fish* OR shellfish OR marine OR freshwater OR carp OR catfish OR prawn OR salmon OR shrimp OR tilapia OR trout) AND TITLE-ABS-KEY (Asia OR "Southeast Asia" OR "South Asia" OR "East Asia" OR mekong OR afghanistan OR "American Samoa" OR bahrain OR bangladesh OR bhutan OR "Brunei Darussalam" OR cambodia OR china OR "Chinese Taipei" OR "Cook Islands" OR "Democratic People's Republic of Korea" OR fiji OR "French Polynesia" OR guam OR "Hong Kong" OR india OR indonesia OR iran OR iraq OR jordan OR kiribati OR korea OR kuwait OR "Lao People's Democratic Republic" OR lao OR laos OR lebanon OR macau OR malaysia OR maldives OR "Marshall Islands" OR micronesia OR mongolia OR myanmar OR nauru OR nepal OR "New Caledonia" OR niue OR "Norfolk Island" OR "Northern Mariana Islands" OR oman OR pakistan OR philippines OR palau OR palestine OR "Papua New Guinea" OR "Pitcairn Islands" OR qatar OR samoa OR "Saudi Arabia" OR singapore OR "Solomon Islands" OR "Sri Lanka" OR "Syrian Arab Republic" OR taiwan OR thailand OR timor-leste OR tokelau OR tonga OR turkey OR tuvalu OR "United Arab Emirates" OR vanuatu OR "Viet Nam" OR vietnam OR "Wallis and Futuna Islands" OR yemen)

CNKI

TI = ('抗生素' + '抗菌' + '兽药' + '兽用药' + '兽用抗生素' + '用药' + '抗微生物') AND TI = (escherichia + (E*coli) + coliform + vibrio + Photobacterium + Aeromonas + Edwardsiella + Edwardsiellosis + Yersinia + Pseudomonas + Flavobacter + Piscirickettsia + Hepatobacter + Francisella + Chlamydia + Mycobacter + Nocardia + Streptococcus + Lactococcus + Aerococcus + Renibacterium + Clostridium + Enterobacterium + '大肠菌' + '埃希菌' + '弧菌' + '光细菌' + '气单胞菌' + '埃德华氏菌' + '耶尔森氏菌' + '假单胞菌' + '黄杆菌' + '立克次氏体' + '肝杆菌' + '弗朗西菌' + '衣原体' + '分枝杆菌' + '诺卡氏菌' + '链球菌' + '乳球菌' + '空气球菌' + '肾菌' + '梭菌' + '肠杆菌') AND TI = ('水产' + '鱼' + '渔' + '贝' + '海' + '淡水' + '鲤' + '虾' + '鲑' + '罗非' + '鳟')) OR (KY = ('抗生素' + '抗菌' + '兽药' + '兽用药' + '兽用抗生

素’ + ‘用药’ + ‘抗微生物’) AND KY = (escherichia + (E*coli) + coliform + vibrio + Photobacterium + Aeromonas + Edwardsiella + Edwardsiellosis + Yersinia + Pseudomonas + Flavobacter + Piscirickettsia + Hepatobacter + Francisella + Chlamydia + Mycobacter + Nocardia + Streptococcus + Lactococcus + Aerococcus + Renibacterium + Clostridium + Enterobacterium + ‘大肠菌’ + ‘埃希菌’ + ‘弧菌’ + ‘光细菌’ + ‘气单胞菌’ + ‘埃德华氏菌’ + ‘耶尔森氏菌’ + ‘假单胞菌’ + ‘黄杆菌’ + ‘立克次氏体’ + ‘肝杆菌’ + ‘弗朗西菌’ + ‘衣原体’ + ‘分枝杆菌’ + ‘诺卡氏菌’ + ‘链球菌’ + ‘乳球菌’ + ‘空气球菌’ + ‘肾菌’ + ‘梭菌’ + ‘肠杆菌’) AND KY = (‘水产’ + ‘鱼’ + ‘渔’ + ‘贝’ + ‘海’ + ‘淡水’ + ‘鲤’ + ‘虾’ + ‘鲑’ + ‘罗非’ + ‘鳟’)) OR (AB = (‘抗生素’ + ‘抗菌’ + ‘兽药’ + ‘兽用药’ + ‘兽用抗生素’ + ‘用药’ + ‘抗微生物’) AND AB = (escherichia + (E*coli) + coliform + vibrio + Photobacterium + Aeromonas + Edwardsiella + Edwardsiellosis + Yersinia + Pseudomonas + Flavobacter + Piscirickettsia + Hepatobacter + Francisella + Chlamydia + Mycobacter + Nocardia + Streptococcus + Lactococcus + Aerococcus + Renibacterium + Clostridium + Enterobacterium + ‘大肠菌’ + ‘埃希菌’ + ‘弧菌’ + ‘光细菌’ + ‘气单胞菌’ + ‘埃德华氏菌’ + ‘耶尔森氏菌’ + ‘假单胞菌’ + ‘黄杆菌’ + ‘立克次氏体’ + ‘肝杆菌’ + ‘弗朗西菌’ + ‘衣原体’ + ‘分枝杆菌’ + ‘诺卡氏菌’ + ‘链球菌’ + ‘乳球菌’ + ‘空气球菌’ + ‘肾菌’ + ‘梭菌’ + ‘肠杆菌’) AND AB = (‘水产’ + ‘鱼’ + ‘渔’ + ‘贝’ + ‘海’ + ‘淡水’ + ‘鲤’ + ‘虾’ + ‘鲑’ + ‘罗非’ + ‘鳟’))

Grey literature searches:

AGRIS (<https://agris.fao.org>):

(Resistance OR "antibiotic resistance" OR "antimicrobial resistance") AND (bacteria species) AND (aquaculture OR aquatic OR fish OR shellfish OR marine OR freshwater OR fish spp.) AND (Asia OR "Southeast Asia" OR "South Asia" OR "East Asia" OR Mekong)

CGIAR FISH (<https://fish.cgiar.org/publications>):

Resistance OR "antibiotic resistance" OR "antimicrobial resistance"

IFPRI (<https://www.ifpri.org/publications>):

Resistance OR "antibiotic resistance" OR "antimicrobial resistance"

WorldFish (<https://www.worldfishcenter.org/publications>):

(Resistance OR "antibiotic resistance" OR "antimicrobial resistance") AND (Asia OR "Southeast Asia" OR "South Asia" OR "East Asia" OR Mekong)

In the appendix, the authors described in good details what they have done but the motivation for their decisions was not properly covered. Further, in some instances, reporting of model building process was little confusing. For example in the appendix under the marine protocol sub-section, line 264, the authors stated “an averaged ensemble model was used ...”, without reporting the rationale for using this approach - for example “weighted averaged models” might be a more appropriate choice here as not all the models have equal importance.

In reference to employed ensemble models, the authors should critically discuss the models’ complexity in relation to generalization error of bias and variance. Authors did not discuss the main limitations of their models in relation to reduction of interpretability of complex models, as well as bias and noise considering the source of data.

R1.21 In the freshwater model we use an ensemble method, stacked generalization, in which we train multiple child models and then stack child model predictions for universal kriging. This enables us to harness the predictive power of independent covariates, and capture spatial autocorrelation in distribution of P50. This approach has been shown to improve predictive accuracy over individual model predictions [Bhatt et al. doi:10.1098/rsif.2017.0520].

Reduced interpretability is inherent to all ensembling models. However, the focus of our study is on producing the best possible maps of the current AMR situation rather than risk factor identification. We therefore opted for an ensembling methodology to obtain the best possible predictive performance. We add this to the methods (L468-477):

“Stacked generalization ensemble approaches have been used to model population-level health metrics⁶², and previous work⁶³ has demonstrated that stacking models improves prediction accuracy compared with individual predictive models. The two-step procedure captures both the relationship between P50 and environmental and anthropogenic covariates as well as spatial autocorrelation in the distribution of P50. This approach has recently been used to model the distribution of AMR in terrestrial animals in low- and middle-income countries²⁶. Although there is an inevitable trade off in improved accuracy at the expense of reduced model interpretability, we chose an ensemble approach for prediction accuracy as the focus of our study was to produce the best possible maps of AMR rather than risk factor identification.”

We also add a discussion regarding the selection of child models (L496-504):

“BRT models have demonstrated good predictive performance in handling non-linear relationships and interactions amongst a diverse set of covariates and have been frequently used to model disease distribution^{25,64,67,68}. By generating and combining a collection of models (decision trees) in a sequential stepwise fashion, boosting reduces both bias and variance while protecting against model overfitting. Additionally, BRT models are insensitive to outliers⁶⁴. LASSO regression models—here applied to GLM and GAM—facilitate efficient covariate selection by shrinking some regression coefficients and setting others with minor contributions to zero. These features enable a robust selection of covariates, reducing model complexity and strengthening predictive performance.”

In contrast to the freshwater model, no association between P50 and collected marine covariates could be identified, and we interpolated P50 by x and y coordinates. We initially used an averaged ensemble model without weightings. We appreciate the reviewer's suggestion to employ a weighted ensemble model that captures variability across models. In the revised protocol, we use a root mean square error (RMSE)-weighted ensemble model to assess the fit and variance of each constituent model in the ensemble model. We find that the constituent models perform similarly ($RMSE_{idw} = 0.2359776$; $RMSE_{nn} = 0.2255336$; $RMSE_{ok} = 0.2131811$). We use the RMSE-weighted ensemble model in the final marine map.

We have added context and described the rationale for employing an RMSE-weighted ensemble model to map AMR in marine environments (L537-541 and SI L418-L422).

“These models were subsequently stacked and weighted according to their root mean square error (RMSE) to capture the fit and variance of each model in the final ensemble model. The weights were taken as the inverse of the RMSE of each constituent model divided by the sum of RMSE for all models, and expressed as their relative proportion in the final RMSE-weighted marine AMR ensemble model (Supplementary Table S3).”

Considering reported data in illustrations, some figures and maps lacks proper data elements. For instance, figure 1 in the manuscript could be augmented with information on reported number of surveys and/or isolates per year. A proper color legend should be provided here too.

R1.22 We have changed the graphical representation of Figure 1, adding a scatter plot background displaying each of the individual surveys in each year. This solution also removes the necessity to represent survey density in each year using boxplot color transparency. We thank the reviewer for helping us strengthen the representation of Figure 1.

References that might be used to improve reporting of this study findings

- Wei, D., Tang, K., Wang, Q., Estill, J., Yao, L., Wang, X., Chen, Y. and Yang, K. (2016), The use of GRADE approach in systematic reviews of animal studies. *Journal of Evidence-Based Medicine*, 9: 98-104. <https://doi.org/10.1111/jebm.12198>
- Percie du Sert N, Hurst V, Ahluwalia A, Alam S, Avey MT, et al. (2020) The ARRIVE guidelines 2.0: Updated guidelines for reporting animal research. *PLOS Biology* 18(7): e3000410. <https://doi.org/10.1371/journal.pbio.3000410>

R1.23 We appreciate the reviewer calling our attention to these resources.

Reviewer #2 (Remarks to the Author):

Response to Reviewer #2

We are grateful to the reviewer for their highly valuable remarks contributing additional context for our findings.

This study by Schar et al. sets out to investigate the temporal trends in antimicrobial resistance (AMR) in aquatic animal products in Asia, and it relies on reported data and geospatial modeling.

AMR in the environment is a major threat to human health at global scale, and aquaculture is one of the major drivers, which has been widely studied, but large scale analysis has seldom been done so far. Therefore, it is a timely study, and will provide a bench mark for further surveillance of environmental AMR and help guide policy to mitigate the spread of AMR in animal industry. Overall, I think about 70% of the paper is more towards a meta-analysis, and with about 30% effort in modeling and prediction, which represents the major novelty aspect of the paper. However, I have some major concerns that might need to be addressed before this can be accepted in a board appealing journal-Nature Communications. Let me be specific as below.

1. I think, by using satellite data, one can map out aquaculture facilities/areas in the region, then look at the distribution of the 749-point prevalence surveys on top of the map derived from geospatial information.

R2.1 We conducted a search of remote sensing data repositories (UN FAO Geo-Spatial Data Network [<http://www.fao.org/geonetwork/srv/en/main.home>]; UN FAO Livestock Production Systems [<http://www.fao.org/livestock-systems/global-distributions/en/>]; NASA Earth Observation Data [<https://earthdata.nasa.gov>]; WorldFish and CGIAR Fish) and were unable to identify geospatial data on aquaculture operations in Asia. Although recent advances in satellite data availability and processing methods (Ottinger et al. 10.3390/rs9050440; Kang et al. 10.3390/su11247186) have been described, to the best of our knowledge, fine scale resolution aquaculture maps are not yet available for the vast majority of Asia. For example, Ottinger et al. project aquaculture production for 104,739 km² of coastal China and Vietnam, representing 1.1% of the combined land area of China and Vietnam (9,734,773 km²) and 0.2% of continental Asia (44,615,653 km²). We fully agree with the reviewer regarding the value of aquaculture geospatial data, which could be used as an additional covariate to inform future iterations of our modeling when such data become available.

Further data on volume of production maybe available in governmental statistics year book? It will also allow to analyze the temporal trends in areas and (production perhaps). This information is important is postulating future surveillance foci considering both point data source but also weighted against area/production scale;

R2.2 Thank you. We agree with the reviewer on the value of incorporating aquaculture production volume in our estimates. In the absence of gridded aquaculture production maps, and since these data are not consistently available publicly at sub-national level in Asia, we found that the best solution was to use the FAO FishStat data repository to obtain production volumes. The FishStat data is supplied by governments to FAO and undergoes a further curation and quality assurance process (FishStat data source) [<http://www.fao.org/fishery/statistics/global-aquaculture-production/4/en>]. We use the latest national level freshwater aquaculture production volumes from FishStat as a weighted factor in our protocol for optimizing future surveillance effort (L522 and L551).

2. From the SI information, I can see that in general, among the 749 surveys, more surveys were conducted in more recent years. Should we also discuss the this potential bias? E.g. in 2002

R2.3 We thank the reviewer for highlighting this point, and helping us to clarify how we communicate this potential bias. Whereas the Supplementary Figures S4 and S5 display publication date, our regressions on temporal trends use sampling date reported in the survey to accurately capture the time during which sampling was conducted in each survey. Below is the annual distribution of sampling dates for all surveys used in the regressions on temporal trends:

A scattered temporal distribution is observed in wild caught surveys in particular. We find that when we divide the surveys at the year that most closely partitions each of the cultured and wild caught data sets in half, and analyze the temporal trends for each time series, the directionality of the regressions trends is unchanged.

Annual trends in the proportion of drugs with resistance greater than 50% (P50) in each survey by time series. (a) P50 for cultured aquatic animals before 2011 ($n = 292$; $p=0.169$); **(c)** cultured aquatic animals after 2011 ($n = 266$, $p=0.953$); and **(b)** wild caught aquatic animals before 2010 ($n = 41$, $p=0.171$); **(d)** wild caught aquatic animals after 2010 ($n = 40$, $p=0.615$). 95% confidence intervals are shown in shaded areas.

We have incorporated this into the discussion (L346-351):

“Overall, there was an increasing volume of surveys conducted in recent years (Supplementary Fig. S4). However, when evaluated by time series that most closely partitions each of the cultured and wild caught data sets in half (cultured: 2000 to 2010 vs 2011 to 2018; and wild caught: 2000 to 2009 vs 2010 to 2016), the directionality of the regressions remained unchanged, suggesting the increasing volume of surveys does not influence the temporal trends in P50.”

3. Further to my point 2, is there any assessment regarding the temporal-spatial matching? I guess this information will be useful in discussing the bias and uncertainty in modeling and also general conclusion. Does the coverage of regions correspond to the proportion of the production volume of that region? How does it matter in terms of modeling? I bet it does, particularly in directing future surveillance efforts;

R2.4 Thank you. To the best of our knowledge—and in contrast to terrestrial food producing animal species—no gridded aquatic food animal population dataset exists. The availability of future data at this fine scale resolution could be used to inform our modeling.

There is a positive correlation between survey coverage and production volume in both freshwater and marine aquaculture in our data set.

Surveys conducted versus aquaculture production volume at country level. (a) freshwater aquaculture; and **(b)** marine aquaculture. Survey counts and production volumes are log10 scaled. Pearson’s product moment correlation coefficient is shown with corresponding p value. Production volumes from FishStat, excluding bivalve molluscs and aquatic plants. Countries are represented by their International Organization for Standardization (ISO3) country code.

We fully agree that accounting for variability in production volumes is important, recognizing the potential contribution of the aquaculture industry to AMR and impact of AMR on the industry. We therefore account for this variability by including inland aquaculture production volume as a weighted factor in our protocol for optimizing future surveillance effort (L546-562; Figure 4). That is, areas with higher relative aquaculture production volume have greater priority for future surveillance.

Regarding temporal-spatial matching, we undertook two additional analyses.

First, we re-evaluated our geospatial models, performing a time series split at the mid-point of the study period (2010). We found that the volume of surveys was insufficient to split into multiple time frames while maintaining the same level of predictive accuracy in our geospatial

models. For even the larger freshwater data set ($n=500$), this mid-point time series split resulted in loss of predictive accuracy (mean AUC = .58 compared with AUC = .60 for the complete time series).

Second, as articulated in R2.3, we evaluated the P50 trends by time series to determine if surveys reported earlier or later in the study period displayed different trends, which could suggest temporospatial patterns in the data. We found that the directionality of regressions on temporal trends remained consistent across time series.

Given data constraints, we avoid overextending our models to spatio-temporal frameworks that may lead to spurious findings. However, as more surveys become available, such spatio-temporal modeling may become feasible.

We have highlighted the insufficient volume of surveys precluding spatio-temporal analysis as a limitation in our discussion (L345-346):

“Fourth, the volume of data is currently insufficient to map AMR across different study periods using spatio-temporal interpolation methods.”

Finally, we add new maps of raw uncertainty (kriging variance) and a map of the 95% confidence interval around the freshwater predictions to assist in the interpretation of our findings (Supplementary Figs. S17 and S18.)

4. In figure s8, the information as to antimicrobial use prior to sampling, this is a bit vague and not so robust, how long ago?

R2.5 Figure S8 represents data extracted from point prevalence surveys ($n=63$) in the systematic review that explicitly identified that antimicrobials were (“yes”) or were not (“no”) applied to the sampled animals prior to sampling. Aquaculture antimicrobial use is infrequently reported; our recent systematic review identified 25 studies representing 146 antimicrobial use rates globally between 2000 and 2019 (Schar et al. doi: 10.1038/s41598-020-78849-3).

Because surveys inconsistently report context associated with the positive identification of use, such as drug class and compounds used; dates, frequency and duration of application; and intended purpose—and because these are frequently subject to farmer recall—in this study we extracted only the binary use classification in our pre-defined search and extraction protocol. These data may be found in the raw data set provided as supplementary data 1 (“History AMU”) and accompanying database legend (doi: 10.5281/zenodo.4609884) and are referred to in the discussion (L203-206).

We have augmented the Figure S8 legend for clarity:

“Analysis is limited to surveys that explicitly identified that antimicrobials were (“yes”) or were not (“no”) applied to the sampled animals prior to sampling ($n = 63$).”

5. In figure s9, the correlation between sanitation and prevalence of AMR of wild caught samples, are these sample from freshwater or marine?

R2.6 Figure S9 represents data originating from point prevalence surveys of wild caught animals in both freshwater and marine environments. We have added this clarification to the Figure S9 legend.

In summary, I think this study is of value, and will be an important literature for future studies and surveillance, particularly the compilation of existing survey data in one paper. As I have said earlier, the major strength is geospatial modeling, but I am not a modeler, so I suggest the final decision should weigh towards those reviewers with modeling expertise. In my opinion, if the modeling is robust, the paper can be potential accepted, though it needs much additional data. Also, it may, during the revision, seek collaborator from major countries, as local knowledge of the geography, production and AMR research related to topic (possibly with inclusion of some publications in national language) will increase the robustness of the overall analysis.

R2.7 Thank you for these encouraging comments. Regarding the inclusion of publications in national language, we have conducted our literature search without restriction on language. Records were screened and data extracted from surveys in Chinese, English, Japanese, Korean, Thai, and Turkish.

Additionally, the collected data will be uploaded to resistancebank.org, creating—for the first time—a single repository of AMR point prevalence surveys from aquatic animals conducted in Asia. We expect our work to stimulate submission of additional aquaculture and fisheries-focused surveys, and for our data set to serve as an initial foundation upon which future data will build.

Reviewer #3 (Remarks to the Author):

Response to Reviewer #3

We greatly appreciate the reviewer's comments, which have helped us enhance the clarity and precision with which we communicate our methods and analyses.

This is a nicely written and interesting paper exploring antimicrobial resistance. I have focused my comments on the statistical analyses, which are generally described well.

1. For the linear regression used for temporal trends, what model goodness of fit measures were examined to ensure adequate fit?

R3.1 We used root mean square error (RMSE) to evaluate goodness of fit for the generalized linear model regressions, providing an absolute measure of fit using the units of our response variable (P50). Whereas in the initial analysis, regressions were performed using the median P50 value by year, we have now performed these regressions using all of the individual survey P50 values by year, providing a larger sample of observations with which to fit the regression

models. We present this in a revised Figure 1. The $RMSE_{\text{cultured}} = 0.223$; and $RMSE_{\text{wild caught}} = 0.235$. We thank the reviewer helping us strengthen our approach, which has been added to the revised manuscript in the methods (L439 -L441):

“Root mean square error (RMSE) was used to evaluate goodness of fit for the temporal trends regression models. RMSE indicated model fits were moderate ($RMSE_{\text{cultured}} = 0.223$; and $RMSE_{\text{wild caught}} = 0.235$), consistent with both the scattered nature and scarcity of the data.”

2. Lines 403-406: Please comment on how you checked the assumptions underlying an ANOVA, and whether these assumptions were met.

R3.2 First, we checked normality of distribution on the residuals both visually and with normality tests. We used histograms and q-q plots for visual inspection, and Shapiro-Wilk normality tests. For each ANOVA, we arcsine transformed the P50 values to normalize the distributions of these proportions to satisfy the conditions for the one-way ANOVA tests. Second, we checked and confirmed the homogeneity of variance between groups using a Bartlett test and by examining residual plots. Finally, as the data are comprised of point prevalence surveys conducted across 20 countries in Asia over a span of 18 years, we consider the observations to be independent. We add this detail to the supplementary information methods (SI L300-303):

“One-way ANOVA tests were conducted on arcsine transformed P50 values to normalize the distributions of these proportions. The distributions of the residuals were checked visually using histograms and q-q plots and with the Shapiro-Wilk normality test. Homogeneity of variance between groups was confirmed using a Bartlett test and by examining residual plots.”

3. You might like to mention as a limitation/scope for future work to consider incorporating currents into the analysis, especially given the apparent impact of human/cattle antimicrobial consumption and water contamination.

R3.3 We thank the reviewer for underscoring the importance of hydrology in distribution mechanics of AMR. This is a nascent area of research that holds promise of contributing to enhanced understanding of AMR risk across compartments. We have added this to the revised manuscript (L277–279):

“Indeed, the distribution mechanics of AMR in aquatic systems under differing hydrological conditions is an expanding area of research that holds promise of contributing to enhanced understanding of AMR risk across compartments⁵⁰.”

Supplementary information

4. Line 217: Is it possible to elaborate on/provide a reference for replicating the freshwater dataset 5 times?

R3.4 We implemented a spatial cross-validation procedure aligned to the Asia sub-regions of our study to ensure the observations from the training and evaluation sets were geographically independent. This procedure guards against overfitting and selection of models with poor capacity to predict to new areas (Gilbert M et al. doi: 10.1038/ncomms5116). Our data set is relatively data-constrained. In order to perform this spatially distinct cross-validation, where some folds had relatively few observations, it was necessary to replicate the data set five times to provide additional values in training the BRT model. This enabled the model to reach minimum holdout deviance on all bootstrap runs. The augmentation does not add new information, but rather duplicates the existing data uniformly. This effectively provides a more robust data set upon which the model is trained, without which predictive accuracy as measured by geographically independent k -fold cross validation AUC was diminished (replication: BRT AUC = .60; no replication: BRT AUC = .57).

5. Lines 219-221: Please mention the number of surveys with precise sampling coordinates provided.

R3.5 Precise sampling coordinates were provided for 43 point prevalence surveys. We have added this notation to the revised manuscript (SI L340). For additional context, out of all of the surveys in our study, 338 (45%) were within a geographic uncertainty range of 20km. A total of 63%, 71% and 84% of all surveys were within uncertainty ranges of 40, 60, and 100 km respectively.

Minor

6. Line 307: Suggest not italicising "overlap area" and "neighborhood area"

R3.6 Thank you. We have corrected this in the revised manuscript.

7. For the legends on Figures 3 and 5, suggest changing P50% to P50.

R3.7 Thank you. We have removed “%” in the legend for Figures 3 and 5.

8. Figure S16: Were there no positive associations, or are they simply not visible due to tight ranges?

R3.8 No positive associations were identified. We have added this notation to the figure legend, and have added color to outliers to clearly indicate association. We thank the reviewer for helping us clarify the communication of Figure S16.

REVIEWERS' COMMENTS

Reviewer #2 (Remarks to the Author):

I have now had the opportunity in reading the revised version and the responses to reviewers comment point-by-point, overall, I think I tend to agree that the authors have adequately addressed these concerns, and their arguments are robust. Given the general lack of data over long period and large spatial scale, I believe this paper will serve as a nice baseline paper for future investigation.

In summary I think the paper can be accepted for publication now.

Reviewer #3 (Remarks to the Author):

The authors have addressed my comments satisfactorily.

Reviewer #4 (Remarks to the Author):

I have not done a full review of the manuscript, but focused on reviewing the authors' responses to reviewer #1, as this was the task requested by the editor.

Any points raised by reviewer #1 that I do not particularly address, are points, which I think were answered satisfactorily by the authors.

Also, I could not copy-paste from the rebuttal file (301311_1_rebut_0_qdqvrh.pdf) or the converted one, so I refer to the reply number.

R.1

The reviewer questioned the exclusion of other reviews and meta-analyses. I accept the answer provided by the authors for not including these, but would like to know, if the authors went through the references/studies included in the excluded reviews and meta-analyses to see, if these included any relevant references that were not captured in the search?

R.4

I agree with reviewer #1 that the P50 metric is a quite 'rough' metric. However, if the purpose of the metric as explained by the authors is mainly to map occurrence and trends of multi-drug resistance, I think it is useful. Because of the complexity of pathogen-drug class combinations, I think it would be difficult to find another single metric that would be more useful. The authors do also calculate the mean resistance and the P30 and find agreeing trends supporting the choice of P50.

However, I do not agree with the authors' claim that the P50 is "a useful marker of potential therapeutic efficacy". This is mainly an ecological study and the results can in my opinion not guide or support any decisions on clinical treatment at the individual (farm) level.

R.12

Regarding the geospatial modeling, I think the authors did the best they could with the data at hand. Obviously, imperfect data on geolocation with the consequent need for random assignment can always be discussed, as important geographical/demographic/topographic factors may change over very short distances. This could maybe be elaborated further in the discussion.

Regarding the actual models, I have a few comments.

I might have missed it, but did you compare the accuracy of the base/child models with that of the stacked model? Typically, the stacked model performs better, but it is not always the case-

On p.22, please, specify the environmental and anthropogenic covariates that were included for predictions. I am aware that they are presented in the supplement, but I really think they need to be mentioned in the main text as well.

On p.24, please, explain better the combined marine data set. It is stated that "these surveys were then combined", but which surveys? Some (inland freshwater) were also excluded. I assume it is the surveys with exact geolocation and then the surveys where geolocation was randomly

assigned, but it is not very clear.

R.13

I rather like the "need-for-surveillance index", but agree with the previous reviewer that including human population density holds some assumptions. Aquaculture may be produced in low density areas and then transported/exported to other areas, where the majority of the products are consumed. However, if the assumption is that AMR is primarily spread through the interaction (excluding consumption) between humans, aquaculture farms and the environment, which would be my best guess, the index makes a lot of sense - and the 'consumption transmission route' may actually not be that important.

REVIEWERS' COMMENTS

Reviewer #2 (Remarks to the Author):

I have now had the opportunity in reading the revised version and the responses to reviewers comment point-by-point, overall, I think I tend to agree that the authors have adequately addressed these concerns, and their arguments are robust. Given the general lack of data over long period and large spatial scale, I believe this paper will serve as a nice baseline paper for future investigation.

In summary I think the paper can be accepted for publication now.

R2.1 We thank the reviewer for these supportive comments.

Reviewer #3 (Remarks to the Author):

The authors have addressed my comments satisfactorily.

Reviewer #4 (Remarks to the Author):

I have not done a full review of the manuscript, but focused on reviewing the authors' responses to reviewer #1, as this was the task requested by the editor.

Any points raised by reviewer #1 that I do not particularly address, are points, which I think were answered satisfactorily by the authors.

Also, I could not copy-paste from the rebuttal file (301311_1_rebut_0_qdqvrh.pdf) or the converted one, so I refer to the reply number.

R.1

The reviewer questioned the exclusion of other reviews and meta-analyses. I accept the answer provided by the authors for not including these, but would like to know, if the authors went through the references/studies included in the excluded reviews and meta-analyses to see, if these included any relevant references that were not captured in the search?

R4.1 Thank you. Our pre-defined eligibility criteria specifically excluded reviews and meta-analyses. We identified 15 reviews and one systematic review and meta-analysis (supplementary data 2). We conducted a hand search of references cited in these reviews and these had already been included in our review.

R.4

I agree with reviewer #1 that the P50 metric is a quite 'rough' metric. However, if the purpose of the metric as explained by the authors is mainly to map occurrence and trends of multi-drug

resistance, I think it is useful. Because of the complexity of pathogen-drug class combinations, I think it would be difficult to find another single metric that would be more useful. The authors do also calculate the mean resistance and the P30 and find agreeing trends supporting the choice of P50.

However, I do not agree with the authors' claim that the P50 is "a useful marker of potential therapeutic efficacy". This is mainly an ecological study and the results can in my opinion not guide or support any decisions on clinical treatment at the individual (farm) level.

R4.2 We thank the reviewer for this comment, and have removed the following sentences in the SI (SI L283-287):

"Using a 50% threshold of resistance, the P50 metric serves as a useful marker of potential therapeutic efficacy – namely, the probability that antimicrobial chemotherapy options could be effective. Across surveys in our study, 68.5% (n=513) involved recognized bacterial zoonoses. We therefore assessed that the P50 metric was better suited to communicate our findings in the context of clinical therapeutic efficacy."

We replace it with (SI L284-285):

"We use P50 as an index of multi-drug resistance to document the temporal and geographic trends in resistance in bacteria of aquatic animal origin intended for human consumption."

R.12

Regarding the geospatial modeling, I think the authors did the best they could with the data at hand. Obviously, imperfect data on geolocation with the consequent need for random assignment can always be discussed, as important geographical/demographic/topographic factors may change over very short distances. This could maybe be elaborated further in the discussion.

R4.3 Thank you. We discuss uncertainty in our geospatial modeling (L165-168):

"The interpolation of resistance in this study is associated with uncertainty. Variability in geolocation of surveys, in covariates, and in estimates of resistance contribute to this uncertainty, which is captured with a 95% confidence interval map on P50 predictions (Supplementary Fig. S18)."

And we have added context regarding this geographic uncertainty in the discussion (L365-368):

"Similarly, in the absence of systematic surveillance, reliance on passive surveillance data presents variability in survey coverage, which could introduce bias, as well as uncertainty in survey geolocation, adding potential uncertainty to predictive models (Supplementary Fig. S18)."

Regarding the actual models, I have a few comments.

I might have missed it, but did you compare the accuracy of the base/child models with that of the stacked model? Typically, the stacked model performs better, but it is not always the case-

R4.4 We compare the predictive accuracy of each of the child models using the mean value of the area under the receiver operator curve for all bootstrap runs for each model. The BRT AUC = 0.60, LASSO-GLM AUC = 0.57, and LASSO-GAM AUC = 0.56 (SI L368-370). In the stacked generalization, the predictions from each model are used as covariates in the universal kriging procedure. We use a product of the kriging procedure—standardized kriging variance—to assess spatial interpolation uncertainty and to calculate a 95% confidence interval on the P50 predictions (Supplementary Figs. S17 and S18, respectively). While there is not a goodness of fit metric for universal kriging that would provide an equitable comparison with the child models, the kriging variance and 95% confidence interval maps provide pixel level assessments of the final model’s predictive accuracy.

On p.22, please, specify the environmental and anthropogenic covariates that were included for predictions. I am aware that they are presented in the supplement, but I really think they need to be mentioned in the main text as well.

R4.5 Thank you. We have added the list of covariates to the Methods (L494-498):

“The 13 covariates were: accessibility to cities; gross domestic product; irrigated land percentage; minimum monthly temperature; terrestrial livestock P50; terrestrial livestock antimicrobial use; human population density; and population densities of cattle, pigs raised intensively, pigs raised semi-intensively, pigs raised extensively, chickens raised intensively, and chickens raised extensively (Supplementary Fig. S13 and Table S2).”

On p.24, please, explain better the combined marine data set. It is stated that “these surveys were then combined”, but which surveys? Some (inland freshwater) were also excluded. I assume it is the surveys with exact geolocation and then the surveys where geolocation was randomly assigned, but it is not very clear.

R4.6 We thank the reviewer for this comment helping to strengthen communication of our approach. We clarify the surveys used in the marine data set (L534-540):

“Surveys from inland freshwater sites were excluded. Wild caught marine animals sampled at land based post-harvest sites were randomly assigned coordinates to open ocean within a radius of .54 to 81 nautical miles (1 to 150 km) from their nearest coastal location (Supplementary Fig. S15). The marine data set consisted of two groups of surveys: (i) surveys from animals sampled at land based post-harvest sites randomly assigned to open ocean; and (ii) surveys originating from marine, coastal marine, brackish water, and coastal brackish water sampling locations (Supplementary Note 4).

This marine data set (n=322) was used to produce inverse distance weighted, natural neighbor, and ordinary kriging models.”

R.13

I rather like the “need-for-surveillance index”, but agree with the previous reviewer that including human population density holds some assumptions. Aquaculture may be produced in low density areas and then transported/exported to other areas, where the majority of the products are consumed. However, if the assumption is that AMR is primarily spread through the interaction (excluding consumption) between humans, aquaculture farms and the environment, which would be my best guess, the index makes a lot of sense - and the ‘consumption transmission route’ may actually not be that important.

R4.7 Thank you. The “need for surveillance” index is intended to capture AMR impact on both the aquaculture industry and on human health. Concerning human health impact, we consider all exposure routes as potentially contributing to transmission risk, including consumption and aquatic environmental exposure. Of the top ten countries with the largest volume of aquaculture production (9 of which in Asia), 89% of farmed fish are consumed within the country’s domestic markets (Belton B et al. 10.1016/j.gfs.2017.10.005). For example, in China and India, more than 90% of aquaculture output is consumed domestically. We therefore use human population density to capture all potential human exposure pathways, including predominantly domestic consumption and aquatic environmental exposure and we add the Belton et al. reference accordingly:

“This function weights the necessity for surveillance to locations where the potential exposure risk and impact of AMR is greatest on the aquaculture industry and human health—via local consumption⁷⁰ and the cyclical exchange of resistant bacteria and their determinants across humans, aquaculture, and the aquatic environment.”